

# Greenland Ice Mapping Project: Ice Flow Velocity Variation at sub-monthly to decadal time scales

Ian Joughin[1], Ben E. Smith[1], Ian Howat[2]

[1]Polar Science Center, Applied Physics Lab, University of Washington, 1013 NE 40th St., Seattle, WA 98105-6698, USA.

[2]Byrd Polar and Climate Research Center, Ohio State University, 1090 Carmack Road, Columbus, OH 43210, USA.

*Correspondence to*: I. Joughin (ian@apl.washington.edu)

**Abstract.** We describe several new ice velocity maps produced by the Greenland Ice Sheet Mapping Project (GIMP) using Landsat 8 and Copernicus Sentinel 1A/B data. We then focus on several sites where we analyse these data in conjunction with

earlier data from this project, which extend back to the year 2000. At Jakobshavn Isbrae and Koge Bugt, we find good agreement when comparing results from different sensors. In a change from recent behaviour, Jakobshavn Isbrae began slowing substantially in 2017, with a mid-summer peak that was even slower than some previous winter minimums. Over the last decade, we identify two major slowdown events at Koge Bugt that coincide with short-term advances of the terminus. We also examined populations of glaciers in northwest and southwest Greenland to produce a record of speedup since 2000.

Collectively these glaciers continue to speed up, but there are regional differences in the timing of periods of peak speedup. In addition, we computed trends for much of the southwest margin of the ice sheet where other work has suggested slowing ice flow in response to increased melting. Contrary to the earlier results, we find no evidence for a slowdown distributed over a wide area. Finally, although consistency of the data generally is good through time and across sensors, our analysis indicates substantial differences can arise in regions with high strain rates (e.g., shear margins) where sensor resolution can become a

factor. For applications such as constraining model inversions, users should factor in the impact that the data's resolution has on their results.

## 1   Introduction

As recently as the 1990s (Paterson, 1994), it was assumed that the Greenland Ice Sheet and its outlet glaciers would respond slowly to climate change. Since the satellite record began, largely since the 1990s, it has proved these early assumptions false.

In particular, many glaciers in Greenland have sped up substantially over the last two decades (e.g., Joughin et al., 2010; Moon et al., 2012; Rignot and Kanagaratnam, 2006), including several of Greenland's largest glaciers (Howat et al., 2005; Joughin et al., 2004; Luckman et al., 2006). In addition to the ice sheet's outlet glaciers, slow-flowing areas near its margin speed up and slow down seasonally (e.g., Joughin et al., 2008a; van de Wal et al., 2008; Zwally et al., 2002) in response to surface meltwater making its way to the bed through moulins, which can penetrate ice more than 1-km thick (Das et al., 2008).



Several groups have produced estimates of velocity for Greenland using Synthetic Aperture Radar (SAR) and optical images (Mouginot et al., 2017; Nagler et al., 2015; Rosenau et al., 2015). As part of the work described here, several maps from 2000 onwards have been produced by the Greenland Ice Mapping Project (GIMP) (Joughin et al., 2010; 2017; Moon et al., 2012).

The GIMP maps have made extensive use of SAR data from the European Space Agency's (ESA) ERS 1&2, the Canadian Space Agency's (CSA) RADARSAT 1, the Japanese Space Agency's (JAXA) ALOS-PALSAR, and the German Space Agency's (DLR) TerraSAR-X and TanDEM-X missions.

In late 2014 the European Union's Copernicus program began providing Sentinel 1A data at 12-day intervals suitable for ice-
sheet mapping (Mouginot et al., 2017; Nagler et al., 2015). With the addition of Sentinel 1B to the Copernicus constellation in late 2016, ESA has begun processing and distributing regular 6-day coverage of Greenland's coastal regions. These coastal data are complemented by several cycles of coverage over the interior each winter to provide annual coverage of the entire ice sheet. These data are routinely ingested into the NASA MEaSUREs program's GIMP velocity products, which are freely distributed through the National Snow and Ice Data Center (NSIDC, 2018). In addition to radar data, the maps from 2014
onwards also include Landsat 8 data.

Here we describe the production of new GIMP velocity maps that incorporate the Sentinel 1A/B and Landsat 8 data along with data from other sources. Although we emphasize the new products, we examine these products in the context of the entire GIMP 17-year time series to estimate seasonal- to decadal-scale variability in Greenland. We analyse the continuity of the data
set and evaluate the magnitudes of any systematic differences between data sets produced using different sensors. Finally, we examine changes in ice flow at several locations in Greenland to demonstrate the utility of the time series for understanding processes related to ice-sheet and outlet-glacier dynamics.

## 2 Methods

The GIMP velocity products are derived using speckle-tracking/feature-tracking cross-correlation algorithms applied to pairs
of SAR or Landsat 8 images. In cases where the interferometric phase is available, it also is included in the solution since it improves the resolution and provides greater accuracy for the component of motion directed in the satellite look direction (Joughin et al., 2010). The GIMP velocity products described here are processed using the same core set of algorithms, which have been described extensively elsewhere (Joughin, 2002; Joughin et al., 2010; 2017). As a result, here we focus only on the details of the processing relevant to Sentinel 1A/B data.

Unlike traditional stripmap SAR data used in earlier GIMP products, which are distributed as spatially continuous images, the data from the Interferometric Wide (IW) mode of the Sentinel 1 Terrain Observation with Progressive Scans SAR (TOPSAR)



are distributed as series of overlapping ~82-by-20 km discrete bursts, acquired along three adjacent sub-swaths. Since the bursts are small relative to the scale of the ice sheet, the first step in our processing is to use the GAMMA Interferometric SAR (ISP) package to assemble each set of bursts into a continuous single-look complex (SLC) image with a width of ~250km and a length of several hundred kilometres. Once the SLCs are assembled, we process them in the same way that we would normal

stripmap (i.e., non-bursted) SAR images, using our in-house speckle-tracking algorithms. Although some data acquisitions traverse the full length of Greenland, we typically break such data acquisitions up in to individual pieces of more manageable size (< ~1200 km).

For many early SAR missions, the accuracy of the orbital state vectors was not sufficient to determine the geometric baseline

(difference in position) between the satellite(s) on successive passes. As a result, ground control points of known elevation and speed often are used to solve for the geometric baseline parameters to produce calibrated velocity measurements (Joughin et al., 1996). In principal, the Sentinel 1A/B orbital state vectors are sufficiently accurate to calculate the baseline and other geometric parameters with little or no ground control (Nagler et al., 2015). Since our workflow is adapted to use ground control points, however, we use such points with Sentinel 1 data to maintain consistency with earlier GIMP products. For the regular

6- and 12-day Sentinel coastal acquisitions, our baseline solutions are largely constrained by bedrock points, where we know that the velocity should be zero. For the winter campaigns, however, some swaths are positioned well away from the coasts. In these cases, we use our existing control-point database, which includes balance velocities and GPS measurements from areas where little change in flow speed is expected (Joughin et al., 2017). We augment these control points by extracting SAR-derived point velocity estimates from areas where there is overlap with well constrained coastal data takes acquired at similar

times.

An advantage of using control points is that they offer the potential to improve the baseline solution, although this improvement has declined over time as orbit reconstructions improve with each new sensor. A second advantage is that the control-points may, at least partially, mitigate other, non-positional, sources of error. For example, the baseline solution can partially

compensate for ionospheric path delays, particularly at L-band for which such delays are larger. The potential downside to using control points is that velocity errors at the control points can bias the solution. Our baseline solutions, however, use 100s to 1000s of points to solve for at most six parameters, providing a relatively robust solution. Furthermore, we have carefully culled the control points to avoid introducing biases (Joughin et al., 2017).

The Landsat 8 data are processed using a cross-correlation-based feature tracking algorithm similar to that used by others (Fahnestock et al., 2016; Jeong et al., 2017). Although the Landsat-8 offsets are produced in map-projected coordinates, we use a control-point procedure to fit a simple plane to each velocity component in each image pair to compensate for geo-location errors (Joughin et al., 2017). In the final stage of calculation, corrections are made for the projection-dependent scale distortion.



Once the SAR and Landsat 8 data have been calibrated using control points, all of the data are combined and mosaicked to the final output grid using our velocity-determination algorithms (Joughin, 2002; Joughin et al., 2010; 2017). At each point in the output, the result represents an inverse-error-weighted (i.e., $1/\sigma^2$) average of all viable estimates. As described below, besides

the inverse-error weighting, additional weighting is applied to the data as needed. As part of the velocity estimation procedure, a formal error estimate is produced for each point in the output grid. In general, these errors agree well (within about a factor of 2) with independent estimates of error (Joughin et al., 2017). Although the error estimates are given as 1-sigma values, the actual distribution likely has a heavier tail than that of a Gaussian distribution, but with the same standard deviation. Care should be taken, therefore, in applying standard tests of statistical significance due to the potential for large outliers.

Most prior SAR missions have provided coverage exclusively along ascending or descending orbits. With the Sentinel 1A/B mission, however, the measurements often cover an area from both ascending and descending orbits, particularly during the winter mapping campaigns. In such cases, we apply a surface-parallel flow assumption to determine the velocity solely from ascending and descending range offsets. The advantage to this approach is that it avoids using the noisier, along-track azimuth

offsets. A disadvantage is that range-offset errors tend to be amplified by about a factor of 3 in the north-south direction. Nonetheless, such solutions generally are far less noisy than pure range-azimuth offset estimates, particularly at times when the ionosphere introduces errors in the azimuth component of the speckle-tracked velocity estimates (Gray et al., 2000). In our final solutions, we include both types of estimates, weighted by their respective inverse-error estimates.

The GIMP project produces two types of velocity products. The first type provides "snapshot" velocities calculated from the displacements in a single image pair. The second type provides aggregated estimates, which represent a single time period but are formed by averaging multiple individual estimates. In the latter case, the individual constituent estimates may not uniformly sample the output interval. In these cases, there are various trade-offs that must be considered to produce accurate results without sacrificing too much temporal resolution or introducing large deviations from the nominal time stamp (temporal skew).

Through GIMP, we produce annual (12-month), winter (up to 9-month), quarterly (3-month), and monthly aggregate products. As an example of the temporal-resolution issues that arise in producing such data sets, consider the case of monthly sampling, with time intervals corresponding to each calendar month. If our only source data are 12-day Sentinel 1A pairs, then some pairs will straddle the beginning and end of each month. In this case, we weight each pair by its duration of overlap with the

month. Absent other weighting, this operation is equivalent to linearly interpolating the 12-day time series to each day of the month, and then averaging the result. This operation necessarily degrades the temporal resolution of the product, since interpolation can be represented as the convolution of a low-pass filter with the data. In this example, such degradation would be relatively small.




Another problem with temporal sampling is what to do with data whose temporal resolution is coarser than the sampling interval. In areas where data are plentiful, coarsely-sampled data may be discarded. There are cases, however, where such data may be the only available, or their inclusion might significantly improve accuracy. As a result, we do include some data with coarser temporal resolution than the nominal sampling interval. In particular, we allow inclusion of data collected over a time

span up to ~60% greater than the sampling interval. Weighting is applied to these data (e.g., for a 30-day interval, a 48-day Landsat 8 would be weighted by a factor of 30/48) so they mainly contribute to the solution in cases where there are insufficient finer-resolution data. Thus, in some areas the temporal resolution of the GIMP products is coarser than the posted sampling interval, with the degree of resolution degradation depending on the spatially varying mix of data used to compute the velocity.

Both the weighting to accommodate temporal sampling and the inverse-error weighting can skew the nominal centre date for each aggregated product. For example, if a subset of the data is much less noisy, it will contribute more heavily to the average and skew the effective date toward the time covered by the less noisy result. Similarly, missing data can skew the effective time away from the nominal sampling time. To indicate where a large temporal skew may have occurred, we apply the data weights to compute the average of the deviation of the date for each pair from the nominal centre date. Since this scalar variable

represents a complicated average of many data, it best used to diagnose cases where temporal skew could be an issue rather than to serve as correction. All data plotted here are shown for their nominal time stamp.

While our goal is to produce uniform sampling, some degree of temporal skew and loss of resolution as described above is inevitable. Our approach attempts to optimally balance the level of noise with the size of data gaps for the anticipated common

applications of the data set, as illustrated in the examples below. The individual estimates, however, remain available for cases where more precise timing is needed and a higher noise level can be tolerated (e.g., to compare speedup with terminus retreat events).

## 3    Results

Table 1 summarizes only the GIMP velocity products discussed here. For simplicity it excludes the multi-year average product

(Joughin et al., 2017) and optical-only (Howat, 2017) GIMP products. Of the products in Table 1, the winter velocities and the individual glacier velocities have been described earlier (Joughin et al., 2010; Moon et al., 2012). Here we describe updates to these products and provide details related to several new products.

### 3.1    GIMP Products

The oldest set of GIMP products presented here is a series of winter velocity maps, which extend back to the winter of 2000/1

and are derived entirely from SAR data (Joughin, 2017b). For these maps, we define winter to be the period with little or no melt, extending from September 1 to May 31. Many of these earlier GIMP winter-velocity maps use campaign-mode data and



are hence derived from acquisitions spanning only a few months. The first winter map to include Sentinel 1A data, available beginning in late 2014, was produced largely from data collected toward the latter half of the 2014/2015 winter. The 2015/2016 and 2016/17 winter maps include data from the full, 9-month period along the coasts, with campaign coverage in the interior.

Our individual glacier estimates (see red boxes in Figure 1 for locations) provide 11-, 22-, and 33-day "snapshot" estimates for many of Greenland's fastest outlets, derived using data from DLR's TerraSAR-X and TandDEM-X missions (Joughin et al., 2016a). At present these estimates cover the period from 2009-to-2017, with future coverage dependent on data availability. The high (1–3 m) single-look resolution of these X-band instruments is such that of all of the GIMP products, these products have the best resolution and short-term accuracy.

Regular Sentinl-1A/B and Landsat 8 coverage, now allows GIMP to produce annually-averaged velocity maps from 2015 onwards (Joughin, 2017a). To temporally align these products with other GIMP products, each year extends from December through November (e.g., the 2015 product extends from December 1, 2014 to November 30 2015). Figure 1 shows the 2016 annual velocity map. These annual products tend to have the best coverage because the Landsat 8 data often is able to fill in

those areas, primarily in the high-accumulation areas of the southeast, where SAR methods consistently produce gaps on some glaciers.

The GIMP project also produces a routine set of monthly and quarterly velocity maps as detailed in Table 1. This time series begins in December 2014, and includes Copernicus Sentinel 1A/B, TerraSAR-X/TanDEM-X, and Landsat 8 data. For periods

with daylight, the Landsat 8 data contribute heavily to the results. By contrast, in winter some products are entirely radar derived. For the spring and fall quarterly products in particular, the Landsat 8 data can contribute to the temporal skew because daylight collection is skewed toward the early (fall) or late (spring) part of the period.

### 3.2    Sentinel 1 Coverage and Individual Estimates

The Copernicus Sentinel 1A/B satellites now provide routine coastal sampling, with less frequent sampling in the interior

during each winter. Together, these satellites have mapped the velocity of many coastal areas more than 100 times since late 2014 (Figure 2a), exceeding the collective coastal coverage of all prior SAR missions. A further advantage of the Sentinel instruments is the fine (2.3 m) slant-range sampling of the single-look data, which is second only to the TerraSAR-X/TanDEM-X instruments. The azimuth sampling (13.9 m) of the Sentinel IW TOPS mode is ~7 times coarser, however, than that of TerraSAR-X/TanDEM-X and ~2.7 times coarser than that of RADARSAT 1. Resolution has a direct effect on accuracy, hence

the error in the azimuth  (parallel to the satellite track) component of motion typically is about a factor of 3.5 times worse than that in the range (cross-track) direction (see also Mouginot et al., 2017). In many cases, the azimuth offset accuracy is further degraded by ionospheric effects (Gray et al., 2000).



Successful velocity estimation with speckle-tracking relies on maintaining strong interferometric correlation between images, which declines with the time between image acquisitions due to processes such as melt and firn compaction. As a consequence, the Sentinel 1A/B 6- and 12-day repeat intervals greatly improve the ability to detect displacement relative to the 24-day repeat period of RADARSAT and the 35-day repeat periods of ENVISAT and ERS-1/2 (when not in Tandem or Ice modes). This

improvement is particularly evident for high accumulation coastal areas in the southeast (Figure 2a), where Sentinel 1 provides ~20 or more measurements for regions in which all RADARSAT 1 data, collected over a 13-year period, provided few or no viable estimates (Joughin et al., 2017).

Since displacement data are scaled by the observation interval used to derive velocity, individual Sentinel 1A/B velocities

derived using 6- and 12-day image pairs are relatively sensitive to errors that are uncorrelated in time, such as those caused by the ionosphere. These errors, however, are reduced when stacking (averaging) multiple estimates. To help assess the errors of individual Sentinel 1A/B pairs, Figure 2b shows the combined standard deviation ($\sqrt{\sigma_{v_x}^2 + \sigma_{v_y}^2}$) for data collected beginning in January 2015 and extending through September 2017. For the interior of ice sheet, where there is little or no melt and speeds do not vary significantly, the mean combined standard deviation is 19 m/yr with individual component errors of $\sigma_{v_x}$=6.2 and

$\sigma_{v_y}$=17.5 m/yr. The large difference between the $x$- (east-west) and $y$- (north-south) components indicates the dominance of the azimuth errors, which are more closely aligned with the north-south direction. These values provide estimates of uncertainty for the individual velocity estimates under relatively stable conditions, as evident over ice-free areas in Figure 2b. The data in Figure 2b represent a mix of 6- (~60%), 12- (37%) and 24-day (3%) pairs. As a result, these values should underestimate the uncertainty for 6-day pairs and overestimate it for 12-day or longer pairs.

The causes of the large standard deviations in velocity for coastal areas (reddish areas in Figure 2b) are more complicated because some of the variability indicates actual variations in speed, such as seasonal variability and marine-terminating glacier dynamics. Close inspection of the data, however, reveals that much of the variability, especially on slow moving coastal areas, is due to noise. In these areas, surface melting and other changes (e.g., high firn compaction rates) lead to weaker intra-pair

correlation and, thus, noisier estimates.

Although the results shown in Figure 2b are representative of mean performance, there is substantial variability in data quality, with some estimates being much better than others. For this reason, we expect the temporally aggregated, error-weighted averages described above to yield much lower errors in aggregated velocity maps relative to unweighted averaging. In areas

where there is both ascending and descending coverage, we reduce errors further by including offsets derived entirely from crossing range-offset data as described above.



### 3.3 Resolution and Systematic Differences

Figure 3 shows a comparison of individual Sentinel 1A/B and TerraSAR-X velocity estimates for transverse and longitudinal profiles from Sverdrup Glacier (see Fig. 1 for location) in northwest Greenland. All of these data were collected over a ~3-week period in the summer of 2017. The transverse-profile data (Fig. 3a) indicate summer speedup of about 100 to 200 m/yr

over the period covered by the data. For estimates that are nearly coincident in time, there is agreement between the TerraSAR-X and Sentinel 1A/B data toward the centre of fast flow and in the adjacent, slow-moving areas. Along the shear margins, however, there are systematic differences of up to ~700 m/yr. For the most part, these differences can largely be attributed to differences in the sensor resolution. With an adequate cross-correlation window size and further smoothing to improve accuracy, the effective azimuth-direction resolution is ~1.5 km for Sentinel 1 A/B and 460 m for TerraSAR-X. As a result, the

TerraSAR-X data track the sharp gradients at the shear margins, while Sentinel 1A/B tends to smooth over them.

Figure 3b plots data for a longitudinal profile down Sverdrup Glacier. Over most of the profile, the data agree to within roughly the level of uncertainty indicated by the error bars and any seasonal variation. The error bars represent the random errors for individual estimates, which are uncorrelated from one estimate to the next. Examination of the data, however, indicates some

of the differences between the profiles is sensor-specific rather than random. Some of these differences are attributable to the different resolutions as just described. Other sources of systematic error, however, may be present. For example, the DEM used for the surface parallel flow correction can introduce slope-dependent errors that depend on the imaging geometry (i.e., results from the same sensor with the same viewing geometry have a common error). For past products, we have assumed a worst-case error of about 3% or ~30 to 90 m/yr for the speeds shown in Figure 3, which is of similar magnitude to the observed

differences. For these products, however, we expect improved performance based on the quality of the latest GIMP DEM (Howat, 2017). Because these errors are mixed in with the resolution errors, it is difficult to isolate and quantify each source of error.

We also note that the accuracy of the DEM determines the geolocation accuracy of the final results (~1.25 m horizontal location

error for each 1 m of elevation error). For areas similar to the Sverdrup Glacier example, where thinning rates are small, geolocation errors have a negligible impact on the results. In areas of rapid thinning and strong velocity gradients, such as Jakobshavn Isbrae, the surface height at the time of data acquisition may substantially differ from that of the DEM, resulting in significant geolocation errors. Since the 10's of meters per year variability of Jakobshavn Isbrae make it an extreme case, we have mitigated the resulting errors in TerraSAR-X products using annually updated DEMs.

### 3.4 Outlet Glaciers: Jakobshavn Isbrae and Koge Bugt

As indicated in Table 1, GIMP provides several data sets at a variety of spatial and temporal resolutions generated from multiple sensors, covering overlapping periods. To better understand the temporal consistency of these data sets for outlet





glacier studies, here we examine the speeds over a 9-year period, during which there is substantial overlap of the various data sets. In particular, we focus on Jakobshavn Isbrae and Koge Bugt. Although some inter-comparison for a more limited set of the Jakobshavn Isbrae data has been presented elsewhere (Joughin et al., 2012; Lemos et al., 2018), we present a more comprehensive inter-comparison here. Koge Bugt is estimated to be the glacier with the third greatest imbalance in Greenland

for the period 2000 to 2012 (Enderlin et al., 2014), making its record of flow variability of interest for understanding ice loss. Finally, with high strain rates and as two of the fastest glaciers in Greenland, they represent some of the most difficult areas to map, providing a robust demonstration of GIMP's measurement capabilities.

Figure 4a shows the Jakobshavn Isbrae time series for the last decade plotted using many of the GIMP products listed in Table

1. The labels of the points on the main trunk (see M6, M13, and M20 in inset) give their approximate distance in kilometres from the 2004 terminus, which we use for consistency with earlier work (Joughin et al., 2012; 2008b; 2014). As previously noted, there is a strong annual cycle of speedup, coinciding with summer retreats of the terminus, followed by slowdown during the winter re-advances. A new finding is that the summertime (maximum) speeds have declined since 2012, and the summer 2017 peak is the slowest for the period shown in Figure 4. Moreover, this peak was actually slower than the winter

2016 minimum and just slightly faster than 2015 winter minimum.

To facilitate a more detailed comparison of the data, Figure 4b shows the period from January 2015 to December 2017. The good agreement in this figure between the monthly and 11-day TerraSAR-X/TanDEM-X data largely reflects the dominant contribution of the X-band data to the monthly time series. For the two fastest points (M6 and M13), Sentinel 1A provides no

valid measurements in the early part of the time series. Once Sentinel 1B acquisitions commenced in October 2016, however, the 6-day sampling began providing estimates for the faster flowing ice. At all three points, the Sentinel data provide close agreement with the TerraSAR-X/TanDEM-X data, consistent with other independent Sentinel 1A/B estimates (Lemos et al., 2018).

Figure 5a shows the variation in speed at approximately 3, 6, and 9 km (see KB3, KB6, and KB9 in the inset) from the 2017 terminus position of a large unnamed glacier that discharges ice through Koge Bugt (Bay). In contrast to Jakobshavn Isbrae, there is no clear seasonal signal, although the speed changes substantially on time scales of a few years. From 2007 to 2009, KB3 sped up from just over 6 km/yr to nearly 10 km/yr. For the next few years the speed at KB3 remained relatively constant, before peaking briefly at 11 km/yr in 2012, after which it slowed to ~7.4 km/yr by May 2013. This decline was followed by

another speedup to >10 km/yr by Spring 2015. Speeds then dipped to ~6.8 km/yr in Fall 2016, but picked up again soon after. The prominent signal at KB3 is far more muted at points inland, with speeds 6-km upstream at KB9, staying within the range of 3 to 4 km/yr.





Figure 5b shows same Koge Bugt data over the period from 2015 to 2017. Relative to Jakobshavn Isbrae, the Koge Bugt glacier was sampled far less frequently with TerraSAR-X, so there are several gaps in this time series, especially during the non-summer months. As with Jakobshavn Isbrae, the 12-day Sentinel 1A sampling prior to October 2016 did not yield valid measurements for the two fastest points (KB3 and KB6). Farther inland at KB9, the 12-day Sentinel data worked well, with

almost every possible pair yielding velocity estimates. The missing points in 2015 at KB9 are largely due to the Sentinel 1A acquisition schedule.

In general, the points from the different sensors agree well, although close inspection does indicate that there are some small biases. For example, the summer 2017 speeds at KB9 measured with TerraSAR-X (red triangles) are consistently about 200

m/yr slower than those measured with Sentinel 1 (green circles). Although small, this bias is larger than the expected level of error (< ~100 m/yr). These differences likely result from differing sensor resolution, as described above.

### 3.5   Northwest and Southeast

Numerous glaciers have sped up in Greenland (Joughin et al., 2010; Moon et al., 2012; Rignot and Kanagaratnam, 2006), which is well documented by the GIMP data set. Many of these speedups have been concentrated in northwest and southeast

Greenland. To demonstrate the collective behaviour of northwest glaciers, Figure 6a shows a stack (summed speeds) plot for the 43 glaciers (excluding Jakobshavn Isbrae), for which there is good temporal coverage over the 16-year period. Figure 6b shows a similar plot for 29 glaciers in southeast Greenland.

As shown in Figure 6, from 2000/1 to 2016/17 these glaciers collectively sped up by 38% and 41% for northwest and southeast

Greenland, respectively. Although the degree of regional speedup is similar, its timing differs. In northwest Greenland, the speedup was only 7% from 2000/1 to 2005/6, but there was an increase of 29% over the next 11 years. By contrast, in southeast Greenland the majority of the speedup occurred over the first half of the observation interval (by 29% from 2000/1 to 2008/9). Over the last 8 years, glaciers only sped up by a more modest 9%, mostly due to increases in speed from 2015/6 to 2016/17.

### 3.6   Decadal-Scale Ice Sheet Trends in Southwest Greenland

Earlier work indicated that an area of the land-terminating margin of the Greenland Ice Sheet is slowing down possibly as the result of a more efficient basal drainage network that has evolved to accommodate recent increases in melt (Tedstone et al., 2015). The black rectangles in Figure 7 show the location of this region, which we refer to as T2015. Although the data from that study extend over a much longer interval than the GIMP data, all of the significant change occurred since 2000, which closely matches to the period covered by GIMP. Thus, we use the GIMP data to explore the hypothesis that the changes

observed by the previous study were representative of the behaviour of the southwest sector as a whole. We use the winter data set for this analysis because it provides the longest time span. As noted above, we employ control points in the interior of the ice sheet where we expect change to be small, which could bias the results. To help avoid this problem, we restrict our




analysis to slow-flowing (Figure 7a) areas where the elevation is less than 1400 meters (see white outline Figures 1&7), which we know to be near the coast and thus well constrained by bedrock control points. As Figure 7 indicates, this region roughly corresponds to the bare ice and wet snow zones (Fahnestock et al., 1993), where substantial melt and lake drainage occurs.

Figure 7b shows the spatial distribution (colour) of statistically significant ($p \leq 0.05$) trends in our study area. In the southernmost part of the region, there is a strong speedup trend associated with the large outlet glacier, Narsap Sermia. Similarly, there is a strong positive trend where the top part of the region borders Jakobshavn Isbrae. In the T2015 region (see black rectangle in Figure 7), we find some indication of slowdown, but the trends are less than those estimated by Tedstone *et al.* (2015). One land-terminating glacier displays a pronounced slowdown (SW3 in Figure 7), but the slowing is confined to

near the terminus. Any trends for the rest of the region are small and scattered, and the area with significant slowing is roughly comparable to the 5% of outliers one would expect if there were no trend and *p* were equal to 0.05.

While Figure 7 provides detail about the spatial distribution of trends, it reveals little about the nature of those trends. To provide more detail, Figure 8 shows the full winter time series for four points (NL, SW1, SW2, SW3) shown in Figure 7. The

first point, NL (North Lake) corresponds to an area on the ice sheet were GPS data have been collected over a several-year period (Joughin et al., 2008a; Stevens et al., 2016). While only a few months of each winter are measured by GIMP, the GPS data measure flow over the ~9-month period with little or no melt. Despite this difference in sampling, most of the GPS points agree well with the radar-derived speeds. Although the GPS data suggest a weak trend of -1.3 m/yr$^2$ (p=0.06), the longer radar record reveals no significant trend.

The point SW1 is located in the T2015 region and it has a trend -1.2 m/yr$^2$ (p=0.02), which explains about half of the variance (r$^2$=0.54). Farther to the south at SW2, there is a significant acceleration trend (1.3 m/yr$^2$, r$^2$=0.47). Downstream from this point, there is a slowdown trend at SW 3 (-4.0 m/yr$^2$,), but with a high degree of inter-annual variability (r$^2$=0.37).

## 4    Discussion

We have presented data at several sites around Greenland. Here we discuss the results in the context of overall data quality as well as the processes that contributed to the behaviour at each site.

### 4.1    Data Quality

Overall the times series shown in Figures 4–8 indicate a high level of temporal consistency between data sets. Such consistency is important for tracking and understanding changes in glacier speeds on time scales of weeks to decades. Although Jakobshavn

Isbrae has some of the most rapidly varying seasonal behaviour, the monthly time series captures this variability nearly as well as the 11- and 12-day individual estimates. To the extent that points in the finer resolution time series depart from the monthly





data at both Jakobshavn and Koge Bugt, it is not always clear whether the data reveal actual short-term behaviour (e.g., response to a calving event) or instead are the result of noise.

Relative to the individual snapshot estimates, the monthly time series provide the advantage of greater accuracy through
averaging of multiple estimates. As mentioned above, this accuracy comes at the expense of potential deviation of the actual time stamp from the nominal time stamp. Inspection of the results in Figures 4&5, however, reveals no detectable skew of the monthly data relative to individual estimates. Some of the winter estimates deviate from the more frequently sampled data, but this behaviour is due to averaging of a rapidly varying signal rather than temporal skew in the data. Thus, the data indicate that the trade-off made between accuracy and temporal resolution may often favour the monthly time series for studies of seasonal
variation. Exceptions may occur when trying to study and isolate the effect of specific events (e.g., high melt or calving), where the finer temporal resolution becomes important. Nonetheless, in such cases the observed change needs to stand well above the expected level of uncertainty.

Although the results reveal that the multi-sensor data sets work well for studying glacier changes, Figure 3 reveals some
potential pitfalls for process studies. For example, the smoothed velocity estimates across shear margins from the Sentinel 1A/B data could yield unreliable results when using the data to constrain a model to invert for basal shear stress. Without conducting a detailed sensitivity study, it is difficult to assess how much the results would be degraded using Sentinel 1A/B data instead of TerraSAR-X data in any given application. Qualitatively, however, Figure 3 indicates that if a study needs data that captures high strain rates, then the finer-resolution TerraSAR-X velocities are preferred. This statement applies to a
relatively small fraction (~1% or less) of the ice sheet (i.e., fast outlets). In other areas, the accuracy obtained by averaging large numbers (e.g., Figure 2a) of estimates may weigh in favour of using the results that include Sentinel 1A/B data.

### 4.2    Jakobshavn and Koge Bugt

The much slower 2016/17 winter minimum and 2017 summer maximum seen in Figure 4a indicate a major recent slowdown at Jakobshavn Isbrae. To help examine the cause of this slowdown, Figure 9a shows the history of terminus position plotted
relative to the glacier geometry. The ice thickness of Jakobshavn Isbrae is exceptionally difficult to measure and several maps have been published with substantial differences in bed topography (An et al., 2017; Morlighem et al., 2017), with those differences being especially large along any particular profile along-flow.  Profiles that follow the greatest bed depth for a particular DEM, however, show similar overdeepenings between datasets. Thus, for consistency with past work, we use the Plummer et al. (2008) bed model and caution that the uncertainty in bed topography renders the interpretation somewhat
qualitative as would be the case with other bed DEMs for this region.

Past work has shown that the strong seasonal variations in speed on Jakobshavn Isbrae correspond with retreat and advance of the terminus across a basal over-deepening (Joughin et al., 2012; 2014). Over the course of the seasonal cycle, the speed



increases as the terminus recedes into deep water each summer and slows each winter as it advances into shallower water (see Figure 9a). Peak summer speeds occurred in 2012, when the terminus retreated to the deepest point of the overdeepening. In subsequent summers, the terminus retreated past the overdeepening to shallower water, yielding smaller summer peaks.

In the winter of 2016/17 the terminus advanced nearly 5-km further than any time since the 2009/2010 winter, yielding speeds at least 900 m/yr slower relative to all winter minimums since 2008 (Figure 4). Retreat of the terminus the following summer was also 1–2 kilometres less than the prior 5 summers. Although the late summer of 2017 terminus position was near the point of maximum retreat for 2011, the peak speed in the summer of 2017 was ~1800 m/yr slower than in 2011. The elevation data in Figure 9a, however, reveals that the 2011 terminus was likely at or above flotation as it retreated, resulting in loss of basal

traction during retreat. By contrast, from the surface and bed elevation data we can infer that the ice at the same location was several hundred meters thinner in 2016 and 2017 (orange curves Figure 9a) and floating (see the flotation threshold plotted in blue in Figure 9a), at least while it was located above the overdeepening. Thus, as the terminus retreated in the summer of 2017, it is likely that only contact at the lateral margins, and not the bed, was lost, resulting in a smaller speedup relative to 2011. Moreover, with the thinner terminus in 2017 the pressure boundary condition at ice-ocean interface, which produces a

net driving force on the glacier proportional to the ice thickness squared, would also be substantially diminished, causing a further  reduction in speed relative to 2011 (Howat et al., 2005; Joughin et al., 2012).

It is unclear whether external forcing, internal dynamics, or some combination of both have contributed to the changes in terminus extent that have produced the recent slowdown. Temperature records from the nearby coastal station at Egedesminde

indicate that 2017 was the second coldest year, behind 2015, in the 21st Century (GISS, 2018). Thus, one plausible hypothesis is that the recent colder temperatures may have contributed to the advance and slowdown, although if there were cooler water at the terminus it could have played a role as well. Whether this slowdown could reduce summer thinning and increase winter thickening sufficiently to stabilize the glacier over scales of years to decades is unclear.

Figure 9b shows the variation in terminus position and surface and bed elevation profiles for the Koge Bugt glacier. Unlike at Jakobshavn Isbrae, the bed of this glacier rises above sea level within several kilometres of the terminus (between KB3 and KB6). In the region upstream of approximately 4-km in Figure 9b, the bed is determined using mass-conservation methods constrained by ice-velocity and  radar depth-sounding data  (Morlighem et al., 2017). Downstream of the terminus, bed depths are far more uncertain due to limited availability of bathymetric data. The terminus position data show good correspondence

with the speed record (Figure 5), with slow speeds corresponding to times when the terminus advanced. With the uncertain nature of the bed in the region over which this advance occurred, the terminus could have been grounded or floating. In some of the images used to digitize the terminus position, large tabular icebergs were present, suggesting that, at least at times, the terminus was at or near flotation. Whether grounded or floating, it likely is that the extra resistance provided by the terminus advance produced the slower periods of flow, similar to the case for Jakobshavn. The response at 6 kilometres inland (KB9)



of KB3 is far more muted than the response a similar distance inland on Jakobshavn Isbrae. This difference in behaviour is likely due to the much thinner ice at Koge Bugt, since the distance a stress perturbation at the terminus is transmitted upstream should scale with ice thickness (Cuffey and Paterson, 2010).

With the exception of its brief advances, the Koge Bugt terminus maintained a relatively fixed position over nearly a decade, apparently near the top of an overdeepening. Although this glacier was losing mass rapidly between 2000 and 2012 (Enderlin et al., 2014), it seems unlikely that there could have been strong thinning near the terminus over  period since 2009. Any sustained thinning likely would have caused the terminus to retreat down the reverse slope to the higher ground on the other side of the overdeepening, from which point retreat would have been slowed or stopped by the forward slope and elevations
above sea level. The apparent loss measured by Enderlin et al. (2014) was computed as a discharge anomaly relative to 2000, at which time speeds where similar to the minimums seen in Figure 4. Thus, if the minimum in 2000 represents an anomalously slow period when the glacier might even have been gaining mass, then the Koge Bugt glacier may be losing mass far less quickly than previously indicated (Enderlin et al., 2014), which is consistent with the terminus position data in Figure 9.

It is interesting to compare Koge Bugt and Jakobshavn Isbrae and their relation to their respective topographic settings. If the recent slowdown is not the beginning of a period of stabilization, then the terminus of Jakobshavn Isbrae likely will continue to retreat at least 60 km inland until it recedes from the trough's deeper parts (Joughin et al., 2012). Once this retreat occurs, the terminus would be in a position more like that of Koge Bugt, which has almost completely pulled back out of its trough. Yet the speeds of both glaciers are similar in magnitude. In the case of Koge Bugt, the glacier is able to maintain high slopes
and driving stresses to produce the fast speed necessary to drain the high accumulation along the southeast coast, despite the fact that much of this flow is over a bed well above sea level. This behaviour indicates that once the terminus of Jakobshavn Isbrae reaches the shallower part of its trough, it too may be able to maintain a similar equilibrium.

If climate conditions similar to present persisted over long periods (many millennia integrated over multiple glacial cycles),
then the Koge Bugt terminus may have stayed near its current position as it has over the past decade. This stability likely would occur when sea level was within several meters of present (e.g., other interglacials), because its terminus could not retreat past this point and still maintain contact with the ocean, which would be required in order to evacuate the large volume of snowfall (retreat from the ocean would cause thickening and re-advance). As a result of maintaining its terminus at or near this position for extended periods, the Koge Bugt glacier likely would have caused the erosion that produced the abrupt transition from a
bed above sea level to a deep submarine trough. Initially the head of the trough may have been located farther seaward than present, so that the trough formation would have occurred as its head migrated inland over time. If so, then abrupt transitions at the heads of many subglacial troughs are likely due to a combination of climate and geography causing termini to maintain stable positions at heads of their respective troughs over extended periods. Such points likely represent the "last stands" of these glaciers before a warming climate draws down the ice sheet sufficiently to pull them completely from their troughs.





### 4.3 Regional Outlet Glacier Changes

Figure 6 indicates collective speedups of glaciers in northwest (38%) and southeast (41%) Greenland since 2000, but at rates varying with time. In northwest Greenland, the speedup was greatest over roughly the last 11 years, whereas in the southeast it was the highest from 2001 through 2009. Because the glaciers that sped up have also likely thinned substantially and have

different widths, these results cannot be directly scaled to estimate increased discharge. The patterns of speedup, however, are consistent with estimates of discharge through 2012 (Enderlin et al., 2014) and suggest that discharge has increased since then. In particular, speeds in northwest Greenland have increased by nearly 10% since 2012. Jakobshavn Isbrae was not included in Figure 6, so its recent slowdown could offset some of the northwest discharge increases.

The data from both regions indicate that for individual glaciers there is substantial variability, with some glaciers slowing and other glaciers speeding up in any given year as several earlier results have shown (e.g., Moon et al., 2012). For example, many of the southeast glaciers that sped up through 2008 slowed over the period from 2008 to 2010 and then sped up slightly thereafter. The response of an individual glacier depends on its internal dynamics (e.g., geometry, terminus position, bed conditions) and it its recent history. Thus, apparently similar glaciers subjected to similar forcing may exhibit substantially

different responses, making it difficult to determine the influence of the forcing. With populations of glaciers such as shown in Figure 6, however, it should be easier to determine average response to climate forcing once the records are sufficiently long. Past studies have been hindered by the limited duration of the satellite record, but GIMP and other projects are now producing records of sufficiently long a period to begin such multi-decadal analyses.

### 4.4 Southwest Greenland Ice Sheet Trends

In southwest Greenland, Figure 7 indicates little in the way of significant multi-annual trends in speed for the bare-ice and wet-snow zones over the winters from 2000/01 to 2016/17. Where statistically significant trends occur, they generally are associated with areas of focused outlet flow at either marine or land-terminating glaciers. An exception is a region (~centred on SW1 in Figure 7) that lies within the T2015 region, but with a far smaller areal extent.

Since the Tedstone *et al.* (2015) data set ends in 2014, while ours extend through the winter of 2016/17, differences in observation period may explain some of this difference. Figure 8 indicates that speeds were slower in the winter of 2012/13 after strong melt in the 2012. Thus, if we only compute trends through 2013, the area of significant slowdown surrounding SW1 expands (not shown) to include the area around NL. The spatial patterns of change, however, differ greatly. In our data, the slowdown is concentrated at middle-elevations of the bare ice zone (e.g., SW1), whereas the slowdown observed by

Tedstone *et al.* (2015) is more concentrated near the margin (elevation < ~800 m).





Tedstone *et al.* (2015) use annual velocities rather than winter velocities, as we have used. Each type of data suffers from sampling problems. Our data do not uniformly sample the winter period as described above. They do, however, sample a period when seasonal variation should be minimal. In order to examine the sensitivity to the inconsistent sampling in our data, we compare our data at NL with the GPS data (see Figure 8), which uniformly sampled the full winter (no-melt) period (Stevens

et al., 2016). In the period of overlap, the SAR observations show good agreement with GPS data, suggesting our results are not unduly biased by seasonal variability. The main sampling issue with the T2015 data is that the image pairs they used span a range of values (352 to 400 days) that could alias seasonal variability (e.g., a period of longer than a year could sample fast flow in the summer disproportionately). Thus, some of the observed differences between our results could be due to sampling issues in one or both of the data sets.

If the processes that contribute to the T2015 slowdown occurred entirely in the summer, they would not be detected by our winter data, potentially causing the difference between our winter and the T2015 annual velocities. If the entire slowdown occurred from June through August, then a summer slowdown trend of -6 m/yr$^2$ would be required to produce the annually averaged trend of -1.5 m/yr$^2$ found in the T2015 data. Over a period of several years, such a trend would yield summer

velocities that were slower than winter velocities, which has not been observed thus far. Moreover, when Stevens *et al.* (2016) examined the NL GPS data, they found significant winter slowdown (-1.13 m/yr$^2$) but no significant trend for the summer. Thus, it is difficult to explain the difference between the winter and annual velocities as being largely the result of changes in speed confined to summer periods.

Although we measure winter rather than annual velocities, our data, do not lend support to Tedstone *et al.* (2015) hypothesis that increased melt has caused changes to the basal hydrological system, leading to a sustained slowdown over much of the ice sheet. In particular, we see no evidence for such slowdown in the 325-km stretch of ice sheet to the south of the T2015 region, where similar response to melt would be expected to have occurred. Thus, the trends Tedstone et *al.* (2015) observe may be statistical artefacts, resulting from some combination noise and a shorter-duration (after 2000) record

**5    Conclusions**

By analysing results from new and earlier GIMP products, we demonstrate a 17-year and growing record of temporally consistent ice-sheet velocity data. The varying mix of sensors through time introduces some differences in spatial resolution, which should be considered in any analysis that could be affected. Early results in the time series were derived from only a few image pairs, and for some years there are no data. Over time as TerraSAR-X, TandDEM-X, Landsat 8, Sentinel 1A/B

have come online, temporal sampling and accuracy have improved greatly. Several other SARs are scheduled for launch in the next decade. In particular, the NASA ISRO (Indian Space Agency) SAR (NISAR) is scheduled for launch in 2021. It will sample all of areas of the ice sheet at least 66 times per year (33 cycles each from ascending and descending orbits) with 12-

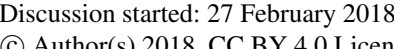



day sampling. Its L-band frequency will improve correlation for difficult to map areas, such as southeast Greenland. Collectively the data from the global constellation will allow GIMP and other products to steadily improve with time. The growing duration of these records will also allow more robust analyses of the processes controlling fast flow and how they are affected by climate and other forcing.

## 6  Acknowledgements

The data described in this paper were produced with support from the NASA MEaSUREs program (NASA grants NNX08AL98A and NNX13AI21A). The RADARSAT data were acquired by the Canadian Space Agency (CSA), and the PALSAR data were acquired by the Japanese Space Agency (JAXA). The German Space Agency (DLR) acquired, processed and distributed the TerraSAR-X and TanDEM-X data), and we gratefully acknowledge the help with these data provided by Dana Floricioiu at DLR. The Sentinel 1A/B Data were acquired through the European Commission's Copernicus program and were processed by the European Space Agency. The RADARSAT data through 2010, the ALOS-PALSAR data, and the Copernicus Sentinel 1A/B data were archived and delivered by the Alaska Satellite Facility (ASF). The 2012/13 RADARSAT data were archived and delivered by CSA. The velocity data sets are distributed through the GIMP project page at NSIDC (http://nsidc.org/data/measures/gimp). All source data are available from their respective space agency and/or ASF. The Landsat 8 data were provided by through a joint effort by the United States Geological Survey (USGS) and NASA and distributed via Google. The NL GPS data were collected as part of a collaboration with S. Das, M. Behn, and L. Stevens. Comments by M. Maki helped improve the manuscript.

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



| Product | DOI (http://dx.doi.org/) | Date Range | Sensors |
|---------|--------------------------|------------|---------|
| Individual Glacier Velocities | /10.5067/MEASURES/CRYOSPHERE/nsidc-0481.001 | 2009-present | TerraSAR-X and TanDEM-X |
| Winter Velocity Maps (September-May) | /10.5067/OC7B04ZM9G6Q | 2000/2001, 2005-2010, 2012/13, 2014-present | RADARSAT 1, ALOS PALSAR, TerraSAR-X, TanDEM-X, and Sentinel 1A/B |
| Annual Velocity | http://dx.doi.org/10.5067/OBXCG75U7540 | Late 2014-present | TerraSAR-X, TanDEM-X, Sentinel 1A/B, Landsat 8 |
| Quarterly Velocity (Nov-Feb, Mar-May, Jun-Aug, Sept-Nov) | https://doi.org/10.5067/1Q1AM4U8Y892 | Late 2014-present | TerraSAR-X, TanDEM-X, Sentinel 1A/B, Landsat 8 |
| Monthly Velocity (by calendar month) | https://doi.org/10.5067/OPFQ9QDEUFFY | Late 2014-present | TerraSAR-X, TanDEM-X, Sentinel 1A/B, Landsat 8 |
| Individual Landsat-8 and Sentinel 1A/B | Delivery to archive scheduled for Fall 2018 | Late 2014-present | Sentinel 1A/B and Landsat 8 |

Table 1. Summary of GIMP velocity data sets and current archival status.



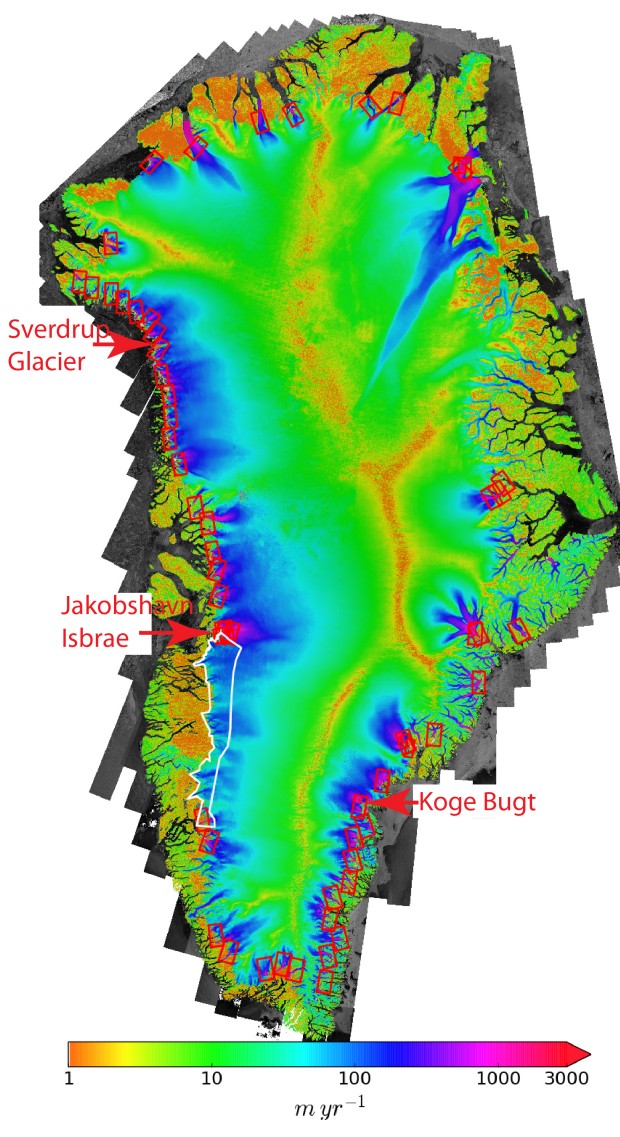

**Figure 1. Annual velocity map for 2016 plotted over SAR image mosaic. Arrows indicate glaciers plotted in subsequent figures. The white outline shows the area shown in Figure 7. Red boxes show the locations of TerraSAR-X and TanDEM-X scenes used in GIMP velocity maps.**





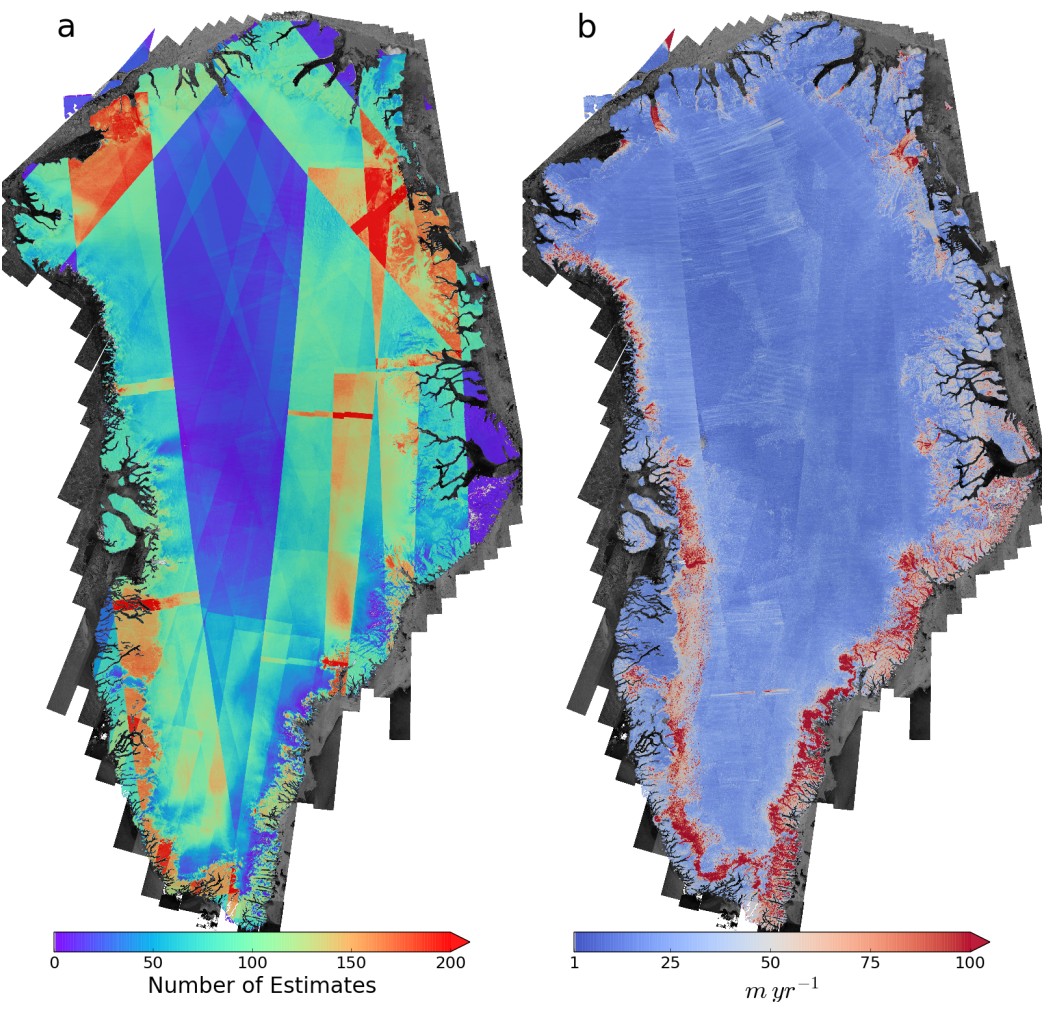

**Figure 2. (a) Number of valid Sentinel 1A/B estimates made using data collected over the period from January 2015 to September 2017. (b) Combined standard deviation ($\sqrt{\sigma_{v_x}^2 + \sigma_{v_y}^2}$) of velocity estimates collected over the same period.**



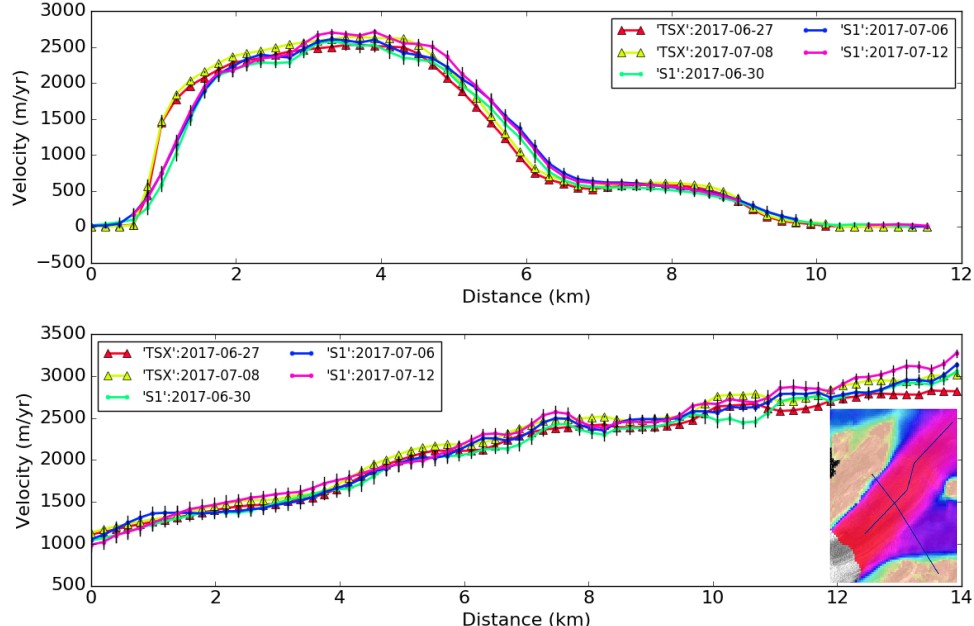

**Figure 3. Profiles of speed from Sverdrup Glacier (see Figure 1 for location) along the (a) transverse and (b) longitudinal profiles shown in the inset. The speeds were calculated using individual TerraSAR-X (TSX) and Sentinel 1A/B (S1) pairs collected in early summer 2017.**



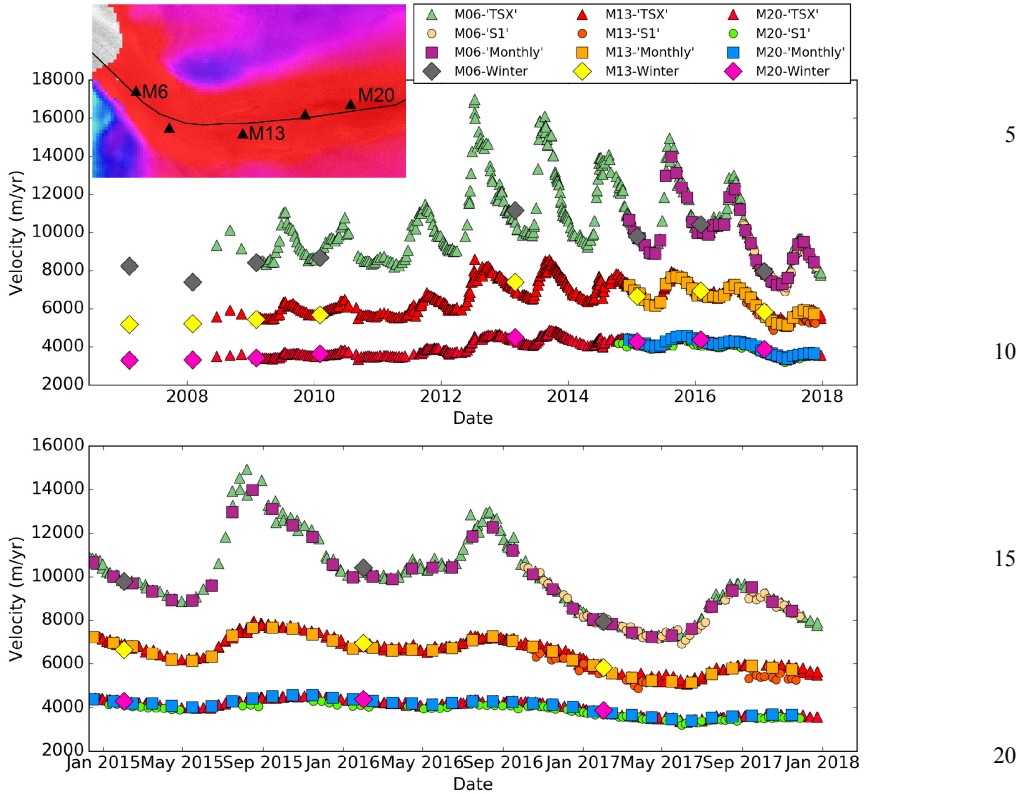

**Figure 4. Speeds for Jakobshavn Isbrae at the points M6, M13, M20 shown in the inset over the periods from (a) 2007 to 2017 and (b) January 2015 to December 2017. Results show individual TerraSAR-X/TanDEM-X (triangles) and Sentinel 1A/B estimates (circles). Also shown are the aggregate monthly (squares) and winter (diamonds) products, which take advantage of all available data. For clarity we omitted the annual and quarterly products listed in Table 1. Error bars also are omitted because they are small relative to the plot marker size.**



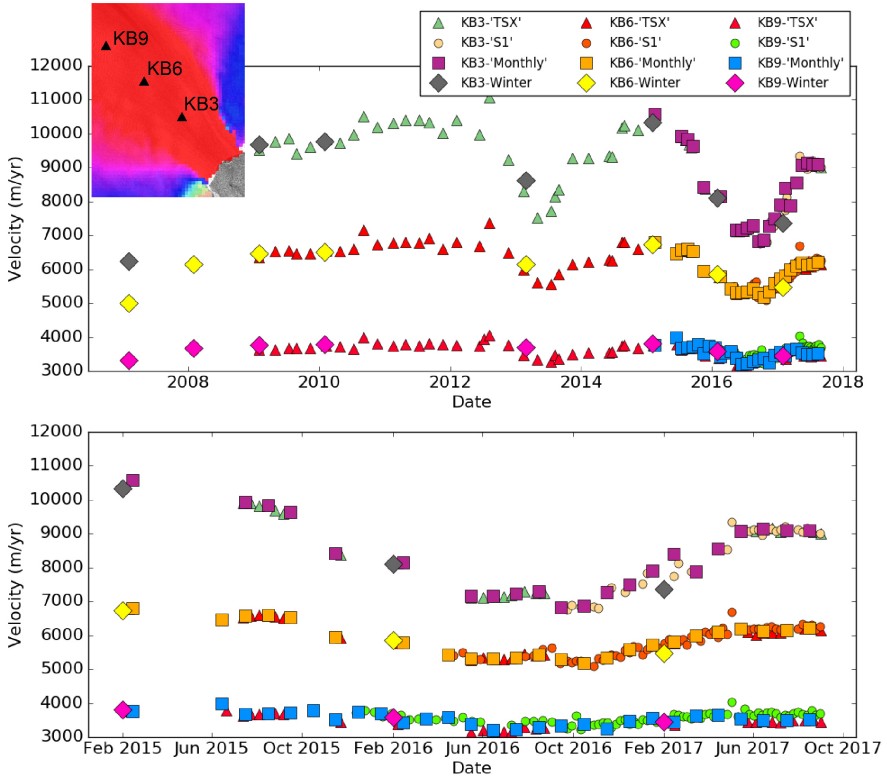

**Figure 5. Speeds for the glacier discharging ice through to Koge Bugt at the points M6, M13, M20 shown in the inset over the periods from (a) 2007 to 2017 and (b) January 2015 to December 2017. Results show individual TerraSAR-X/TanDEM-X (triangles) and Sentinel 1A/B estimates (circles). Also shown are the aggregate monthly (squares) and winter (diamonds) products, which take advantage of all available data. For clarity we omitted the annual and quarterly products listed in Table 1. Error bars also are omitted because they are small relative to the plot marker size.**





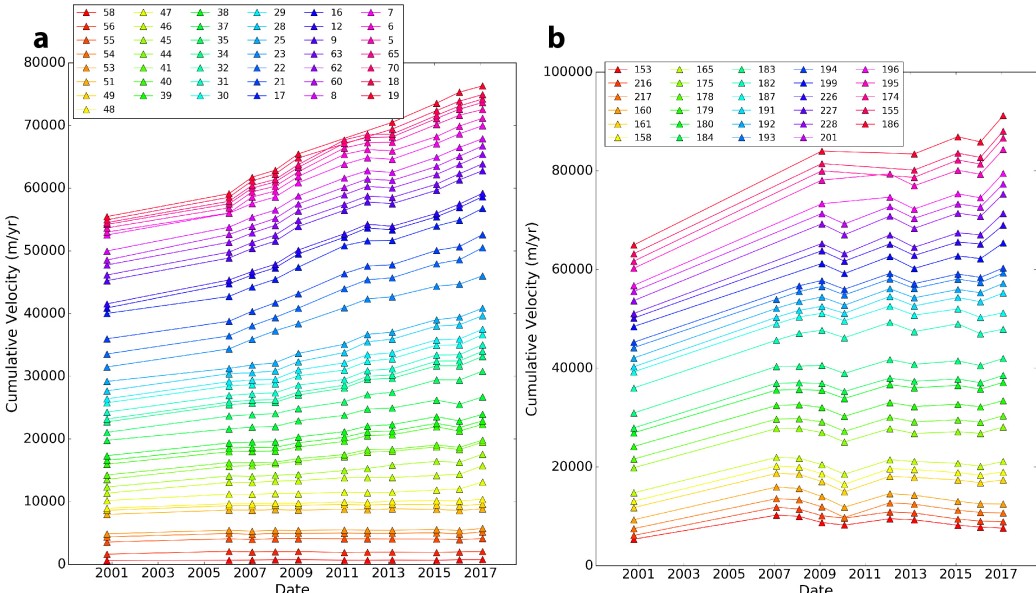

**Figure 6. Summed speeds for collections of glaciers in (a) northwest and (b) southeast Greenland. The legend identifies individual glaciers using the glacier ids (numbers) used for the GIMP terminus position dataset (Moon and Joughin, 2008; Moon et al., 2015). Figure S1 shows the location of each numbered glacier. Because Jakobshavn represents such a large signal (Figure 3), we did not include it the northwest data. For a few glaciers, data is missing for some of the years plotted. In these cases, the stack is arranged with missing points on top so that data on either side bridge the gap.**



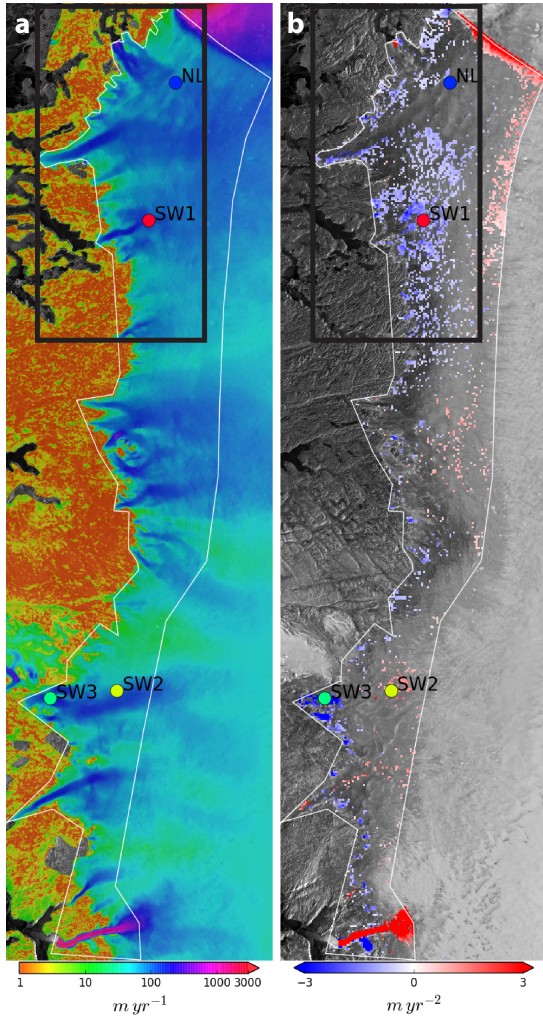

**Figure 7. (a) Speed along a section of the southwest ice-sheet margin and (b) significant (p < 0.05) trends (color) for the period from winter 2000/1 to 2016/2017 calculated for the area enclosed by the white outline. Within the outline, gray indicates no significant trend (p ≥ 0.05). The underlying gray-scale image is a SAR image from RADARSAT (Joughin et al., 2016b). Bright radar returns upstream of the outline generally indicate percolation facies, while darker tones within the outline indicate bare ice or wet-snow facies (Fahnestock et al., 1993). The circles (SW1-3, and NL) indicate the locations for data plotted in Figure 8. The black rectangle shows the approximate area examined by Tedstone et al. (2015), which we refer to as T2015 in the text.**



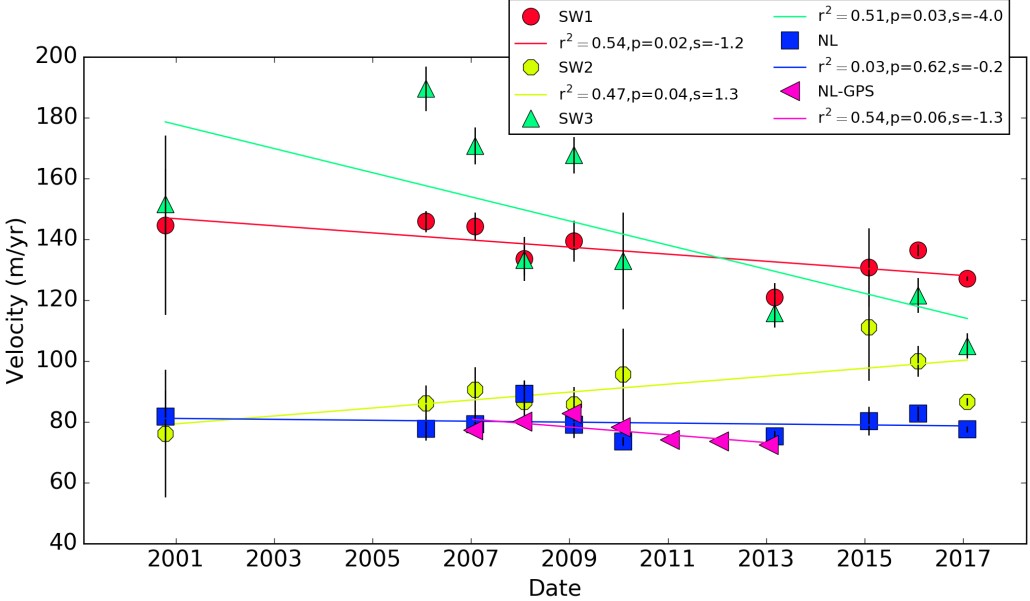

**Figure 8. Winter velocities and corresponding trends for the points SW1–3 and NL (see Fig. 7 for location). Also shown are GPS derived speeds at NL (Stevens et al., 2016).**





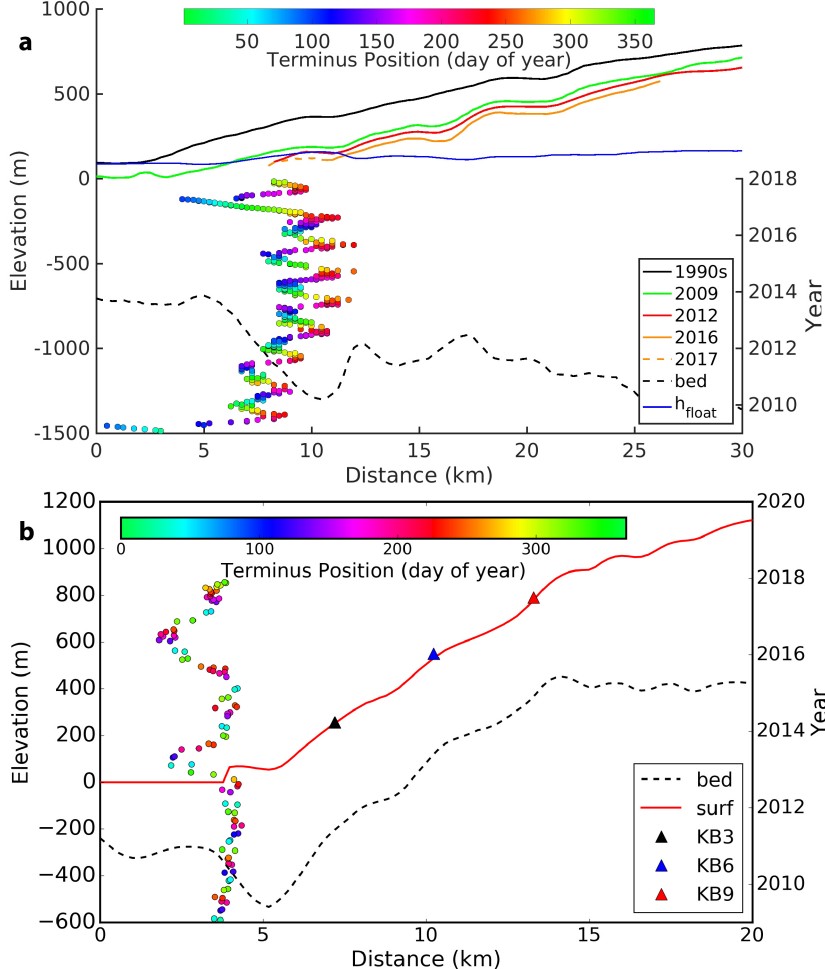

Figure 9. (a) Surface and bed elevation, terminus (2009–2017), and flotation height for Jakobshavn Isbrae updated from a similar figure (Joughin et al., 2014). The bed data are from CRESIS (Plummer, 2008) and the surface elevations are from the NASA Airborne Topographic Mapper (ATM) (Krabill et al., 2004) and WorldView DEMs (Noh and Howat, 2015). (b) Surface and bed elevation and terminus position for Koge Bugt. The surface elevations are from the GIMP DEM (Howat, 2017) and the bed elevations are from Bedmap 3 (Morlighem et al., 2017).