# Peer review of "Greenland Ice Mapping Project: Ice Flow Velocity Variation at submonthly to decadal time scales"

_The Cryosphere, 2018_

## Short Comment (SC1) · 6 Apr 2018

**Comment on 'Greenland Ice Mapping Project: Ice Flow Velocity Variation at sub-monthly to decadal time scales' by Joughin, Smith & Howat.**

A. Tedstone[1], P. Nienow[2], A. Dehecq[3] and N. Gourmelen[2]
[1] Bristol Glaciology Centre, University of Bristol, UK (previously at [2])
[2] School of GeoSciences, University of Edinburgh, UK
[3] NASA Jet Propulsion Laboratory, USA (previously at [2] and Université Savoie Mont-Blanc)

We have read the above discussion paper with interest; the paper demonstrates the tremendous improvements and additions to the GIMP data archive and their potential to enhance records of ice sheet dynamics and the processes controlling dynamic change. However, we wish to raise some concerns regarding the results and discussion of inter-annual velocity trends in the south-west sector of the ice sheet, much of which arises from comparison with our own study on the same topic (Tedstone, Nienow, Gourmelen, Dehecq, Goldberg and Hanna, 'Decadal slowdown of a land-terminating sector of the Greenland Ice Sheet despite warming', *Nature*, 2015; hereafter T2015).

On the south-west GrIS sector, Joughin et al (hereafter J2018) conclude that 'the trends Tedstone et al (2015) observe may be statistical artefacts, resulting from some combination noise [sic] and a shorter-duration (after 2000) record' (P16,L23-24). Broadly, we suggest that, rather than the results from the two studies disagreeing, the differences are likely due to methodological differences in the derivation of the data-sets used and that there are flaws with the current methodology used by J2018 to derive their 'winter' velocity time series. As such, we believe that the broad conclusions from this section of the paper are not currently robust and the specific conclusion that the results from T2015 'may be statistical artefacts' is not justified based on the data presented and should therefore be removed unless considerable further evidence is presented to back up this assertion, including the explicit details of the derivation of the 'winter' time series.

In explaining our concerns in more detail, the rest of this comment is in two parts: (1) an examination of the methodological differences used to derive the respective data sets, and (2) a comparison of the results presented.

**(1) Potential methodological differences**

T2015 mapped the decadal trend in ice motion (their Fig. 1) by differencing two multi-year time periods: 1985-1994 and 2007-2014. Each of these multi-year periods was computed from annual feature-tracked image pairs with a baseline of 352-400 days. In contrast, J2018 use 'winter' velocity mosaics (dataset: NSIDC-0478), which are available for the winters of 2000-01, 2005-06 and 2006-07 onwards; note here that 'winter' is assumed to be any data collected in the nine months from Sept through May and is not uniformly sampled. We understand that the mosaics preceding 2014-15 are predominantly composed of InSAR Campaign-mode data, which was only acquired for a subset of the 9-month winter period. Whilst J2018 have treated these winter mosaics as indicative of net winter ice flow, previous studies show that ice flow varies considerably through winter, which we shall now expand upon.

Variable ice velocity during winter becomes a substantial issue once the degree of variability is considered and if one is trying to characterise a net winter velocity from a temporal subset of winter values. Detailed GPS data presented in Joughin et al (2008, their Fig. 2) show winter velocities increasing from ~55 to ~80 (GPS $V_{NS}$) and ~70 to >95 (GPS $V_{SS2}$) m/yr between September and April respectively; this overwinter velocity change thus represents a ~45% and ~35% increase in velocity between the early winter minimum and late winter maximum. Examples of the same phenomena are also shown clearly in Colgan et al (2012, J. Glac, Fig. 2), Sole et al. (2013, GRL, Fig. 2) and elsewhere where winter GPS velocity records exist. As such, the precise period in winter

when velocities are sampled will have an enormous impact on any 'winter' time series and subsequent trend analysis undertaken.

J2018 note that "many of these earlier GIMP winter-velocity maps use campaign-mode data and are hence derived from acquisitions spanning only a few months" (P5,L31-P6,L1) but do not take the implications of this in to account in the subsequent analyses. For example, the only time period where J2018 explicitly distinguish the 'sub-winter' period of sampling in the manuscript text is for winter 2014-15 when the 'winter' map was "produced largely from data collected toward the latter half of the 2014/2015 winter" (P6, L3). This sampling period would therefore be expected to produce a 'winter' velocity that is considerably enhanced (>~15-20%?) relative to the actual winter velocity, reducing the likelihood of finding an inter-annual slow-down trend. J2018 does not provide any indication of when the 2001 'winter' time-series was collected, just that "early results in the time series were derived from only a few image pairs" (P16, L28-29), but the failure to capture the full winter velocity ensures that the subsequent trend analysis is flawed, especially given the dependence of the inter-annual trend analysis on this sample point at the very start of the time series five years prior to the next sample.

The sampling issue highlighted above is a more significant problem when considering relatively small absolute changes in velocity. The intra-winter velocity range of ~25-30 m/yr reported in Joughin et al. (2008) (and Colgan et al., 2012) is of the same magnitude as the overall annual velocity decrease reported in T2015; as such, any failure to correctly estimate winter motion in the present study will have considerable implications for a trend analysis.

The data underlying the J2018 trend analysis thus fail to capture net ice flow over annual and longer time scales, instead providing sub-annual snapshots of observation periods which vary in their acquisition time, both in length and period during the winter, from one winter to the next. They therefore have the potential to incorporate considerable variability in each of their derived 'winter' velocities, depending on the precise period of time that was sampled/available to derive their velocity estimate. We ask that the authors provide considerably more detail of their different winter 'snapshots', beyond the existing explanation at P16, L2-3 and without simply directing readers to the underlying NSIDC dataset metadata.

We note that J2018 provide a comparison between their radar derived NL data collected over "only a few months of each winter" and NL-GPS data reported in Stevens et al. (2016, GRL). They conclude that "most of the GPS points agree well with the radar-derived speeds" (P11, L17-18) and subsequently suggest that their "results are not unduly biased by seasonal variability" (P16, L5-6). We estimate, taking the data from Fig. 8., that while 'most' GPS do agree well, some comparisons are poor (e.g. 2008 where the SAR data looks to be ~10 m/yr too high, possibly due to seasonal bias). While the comparison gives confidence that the radar is performing reasonably well in terms of absolute GPS displacement (+/- ~5 m/yr s.d.), such an error can introduce considerable variance in velocity trends when the trends are small in absolute terms. As a result, the data as currently presented do not provide compelling evidence that InSAR is generating 'winter' velocities at the requisite temporal resolution that can ensure that the results "are not unduly biased by seasonal variability", especially when investigating trends in areas where the ice is moving slowly (~ 100 m/yr).

Last, J2018 criticise the T2015 choice of baseline period as opening the potential for seasonal variability to be aliased (J2016, P16, L6-9). However, we note that T2015 investigated this possibility in some detail (see Materials and Methods - 'Impact of varying baseline durations on annual velocity' and Fig. S1). J2018 do not make any reference to this analysis in their critique of T2015. To summarise, T2015 found that longer baseline periods beginning/ending in summer are likely to lead to a small artificial increase in ice motion, which is in the opposite direction to the

decadal slowdown signal that is found and reported in the T2015 study area. In line with the T2015 baseline sensitivity analysis, we therefore ask that the authors demonstrate statistically that their own sub-sampling methodology has not impacted their results. Such an analysis should be robust for the whole SW sector analysed if the current conclusions are to be justified.

**(2) Comparison of the results presented in J2018 and T2015**

In J2018 Fig. 7, the units are metres per year. If the aim of J2018 is to undertake a valid and direct comparison with T2015, the authors should use the same units, namely percentage change relative to a reference period. In principle, we assume this would be 2000-01, although given the issues associated with the 'winter' sampling in 2000-1 (and the large error bars associated with this time period as shown in Fig. 8), this would likely be problematic because this earlier reference period may not be representative of net winter motion.

On P11,L7-9, J2018 states that 'In the T2015 region (see black rectangle in Figure 7), we find some indication of slowdown, but the trends are less than those indicated by Tedstone et al. (2015)'. We request that the authors are more precise and provide, for example, an average velocity and average % change for the region. This will ease comparison with both T2015 and the GPS measurements presented in the study. This is especially important given that J2018 find statistically significant evidence for a slowdown (Fig. 7 and P11, L7-9) but conclude later in the manuscript that it is due to aliasing of seasonal variability (P16, L6-7).

The approach chosen for trend analysis (as opposed to differencing two time periods as per T2015) requires clearer explanation. For example, does the analysis take the formal velocity uncertainty at each pixel each year into account, and do they exclude potential outliers (robust linear regression)? What is the estimated error of the computed trends (i.e. computed from the covariance matrix)? Are the pixels used for computing the trend analysis present in every single mosaic or are some pixels missing some time points? Furthermore, we note that a trend of 0 m/yr is a valid result that should not be excluded, yet it appears that these are not shown due to the filtering applied (Fig. 7 caption).

We also note that the GPS observations presented show good agreement with T2015. We presume that, unlike the velocity estimates obtained from InSAR, the GPS dataset records net winter and annual ice motion as opposed to shorter temporal snapshots. NL-GPS (within the T2015 study area) has a computed slow-down of 1.3 m/yr (p=0.06) over the period 2007-2013 (Fig. 8), compared with the T2015 region-average of 1.5 m/yr during 2002-2014 (T2015 Fig. 2 and text). Meanwhile, J2018 fail to reproduce the GPS trend or T2015 trend with their InSAR observations (Fig. 8, NL timeseries). Similarly, a long-term decrease (1990-2012) in annual ice motion in this region has been measured by GPS – at the K-transect (van de Wal et al., 2015, The Cryosphere).

Last, we note that the discussion about summer ice motion at P16,L11-24 could be improved through stronger grounding in existing hydro-dynamic-coupling literature. For instance, studies such as Sole et al. (2013, GRL) show that summer velocities are faster than winter velocities, so proposing summer slow-down to below winter velocities as a possible explanation for T2015 but then immediately 'disproving' it (P16, L14-15) is confusing and has the potential to mis-lead. Similar datasets and discussion can be found in e.g. lead-authored work by Doyle, Sole, Bartholomew, Tedstone, Hoffman, Stevens. Moreover, given that the discussion makes comparison with T2015, it should also directly address T2015's hypothesis, namely that the processes responsible for the slow-down occur following the cessation of melting, i.e. early winter (e.g. T2015, p694, paragraph 1), *not* during summer.

---

## Author Comment (AC1) · 10 Apr 2018

**We have read the above discussion paper with interest; the paper demonstrates the tremendous improvements and additions to the GIMP data archive and their potential to enhance records of ice sheet dynamics and the processes controlling dynamic change. However, we wish to raise some concerns regarding the results and discussion of inter-annual velocity trends in the south-west sector of the ice sheet, much of which arises from comparison with our own study on the same topic (Tedstone, Nienow, Gourmelen, Dehecq, Goldberg and Hanna, 'Decadal slowdown of a land- terminating sector of the Greenland Ice Sheet despite warming', _Nature_, 2015; hereafter T2015).**

*We appreciate the comments on the data and address the concerns below.*

**On the south-west GrIS sector, Joughin et al (hereafter J2018) conclude that 'the trends Tedstone et al (2015) observe may be statistical artefacts, resulting from some combination noise [sic] and a shorter-duration (after 2000) record' (P16,L23-24). Broadly, we suggest that, rather than the results from the two studies disagreeing, the differences are likely due to methodological differences in the derivation of the data-sets used and that there are flaws with the current methodology used by J2018 to derive their 'winter' velocity time series. As such, we believe that the broad conclusions from this section of the paper are not currently robust and the specific conclusion that the results from T2015 'may be statistical artefacts' is not justified based on the data presented and should therefore be removed unless considerable further evidence is presented to back up this assertion, including the explicit details of the derivation of the 'winter' time series.**

*We address a similar question as T2015, which focuses on whether there is a trend in speed in western Greenland from approximately 2000 onward. In summary, our findings using different data, differing seasonal coverage, and covering a slightly different period show virtually no trend, which contrasts with the results presented by T2015. We have done what we can do rule out methodological differences with our data. We were quite careful not simply say "we are right, they are wrong." Instead we tried to go through the possible differences carefully and try to offer explanations based on our understanding of the data. The operative word in the statement cited above is "may", which according to a the first google definition means "expressing possibility."*

*The error bars on the T2015 data are not particularly small (60 m/yr), so it is not unreasonable that noise could be a factor. We also allow for the extra couple of years in our time series to be a contributing factor. We discuss the annual sampling window length below, which we believe can introduce larger biases that T2015 assume.*

In explaining our concerns in more detail, the rest of this comment is in two parts: (1) an examination of the methodological differences used to derive the respective data sets, and (2) a comparison of the results presented.

**(1) Potential methodological differences**

T2015 mapped the decadal trend in ice motion (their Fig. 1) by differencing two multi-year time periods: 1985-1994 and 2007-2014. Each of these multi-year periods was computed from annual feature-tracked image pairs with a baseline of 352-400 days. In contrast, J2018 use 'winter' velocity mosaics (dataset: NSIDC-0478), which are available for the winters of 2000-01, 2005-06 and 2006- 07 onwards; note here that 'winter' is assumed to be any data collected in the nine months from Sept through May and is not uniformly sampled. We understand that the mosaics preceding 2014- 15 are predominantly composed of InSAR Campaign-mode data, which was only acquired for a subset of the 9-month winter period. Whilst J2018 have treated these winter mosaics as indicative of net winter ice flow, previous studies show that ice flow varies considerably through winter, which we shall now expand upon.

> *We were also careful to express there were sampling issues with both data sets. We have directly compared several winters where the SAR data did not include the full 9 months with full-winter GPS estimates and found good agreement, suggesting relatively little bias due to different sampling periods (see Fig. 8 and further discussion of this issue below).*

Variable ice velocity during winter becomes a substantial issue once the degree of variability is considered and if one is trying to characterise a net winter velocity from a temporal subset of winter values. Detailed GPS data presented in Joughin et al (2008, their Fig. 2) show winter velocities increasing from ~55 to ~80 (GPS $V_{NS}$) and ~70 to >95 (GPS $V_{SS2}$) m/yr between September and April respectively; this overwinter velocity change thus represents a ~45% and ~35% increase in velocity between the early winter minimum and late winter maximum. Examples of the same phenomena are also shown clearly in Colgan et al (2012, J. Glac, Fig. 2), Sole et al. (2013, GRL, Fig. 2) and elsewhere where winter GPS velocity records exist. As such, the precise period in winter when velocities are sampled will have an enormous impact on any 'winter' time series and subsequent trend analysis undertaken.

> *Since we have the raw data, we can be more precise about some of the numbers referenced above. There seems to have been some issue with reading numbers off plots in J2008. The range directly from the data (Sept 1 2006 to May 31 2007) is 69 to 88 at NL, with about 10 m/yr of that rise occurring in September. Over the rest of the winter there is a small increasing trend in velocity. All of the radar data at these sites begins in early October or later and ends in April (i.e., the slowest and fastest months are excluded). For the years where we do have coincident GPS and full 9-month estimates, we have good agreement between the "partial" winter estimates and "full" winter estimates.*

*To evaluate the sensitivity the sensitivity to our winter sampling interval, estimated the winter velocity for the North Lake GPS data from Joughin et al 2008 (a full winter of weekly average velocities).  We used a 3-month window (roughly the length of the radar campaigns) and evaluated the results with the window starting on the first of each month, starting in October, which approximates the various radar campaign periods. The corresponding differences between the 9-month and 3-month averages are:*

     *Oct-Jan :*     *0.3 m/yr*

     *Nov-Feb:*     *1.2 m/yr*

     *Dec-Mar*     *2.0  m/yr*

     *Jan-April*     *2.6 m/yr*

*The last sampling window ends in April since we did not include May data for this region. These biases are much smaller than the worst-case biases mentioned in the comment, because we sample the part of the winter when change in speed are relatively muted, and are largely representative of the full 9-month average.*

*Note the biases just mentioned are with respect to the full 9-month winter averages. If every one of our data sets were collected over the same 3-month period each winter, then the biases would have no effect on the trend since they would be approximately the same each winter. Most of the campaign data are from ~Dec-Feb, so we re-ran our analysis with the Sentinel estimates computed just for the Dec-Feb period, which should more closely match the other campaign data. This change made virtually no difference, other than reducing the number of significant trends detected slightly (possibly due to reduced bias or more noisy data), which is opposite to the effect claimed by the T2015 authors  in this comment. Since the differences are small, we opt to keep the less noisy 9-month Sentinel estimates.*

*T2015 evaluated their sampling strategy using a simulated time series, which involved a simple 2-value model with one value for the summer and one value for the winter.  Had we used same simulation as T2015, there would be no bias due to the constant winter value they used.  To evaluate their sampling scheme with more realistic data, we cyclically (annually) repeated the NL GPS data and evaluated annual velocity estimates with window durations ranging from 350 to 400 days, and window start dates from 150 to 226 days. With this test, we obtained annual velocities ranging from 90.4 to 98 m/yr. In other words, the T2015 sampling scheme applied to these GPS data produces annual estimates of speed that deviates from the true value by a range 3x greater than for our winter estimates (7.6 vs 2.6 m/yr).*

*In either case (T2015 or J2018), these biases would have to occur a consistently increasing or decreasing way to greatly change the trend, because if randomly distributed in time they would simply contribute additional noise. In a worst-case situation, our trends could be off by 0.16 m/yr^2 (2.6 (m/y)/16y) while the T2015 results could have an error of up to 0.54 m/yr^2 (7.6 (m/yr)/14 y).*

*The velocities in Colgan et al are pretty stable during the winter, which wouldn't change our conclusion. Neither would Sole et al data which shows pretty flat velocities in the winter. To the extent there are small spikes, these would be smoothed out by our typical 3 -month window for winter velocity determination.*

**J2018 note that "many of these earlier GIMP winter-velocity maps use campaign-mode data and are hence derived from acquisitions spanning only a few months" (P5,L31-P6,L1) but do not take the implications of this in to account in the subsequent analyses. For example, the only time period where J2018 explicitly distinguish the 'sub-winter' period of sampling in the manuscript text is for winter 2014-15 when the 'winter' map was "produced largely from data collected toward the latter half of the 2014/2015 winter" (P6, L3). This sampling period would therefore be expected to produce a 'winter' velocity that is considerably enhanced (>~15-20%?) relative to the actual winter velocity, reducing the likelihood of finding an inter-annual slow-down trend. J2018 does not provide any indication of when the 2001 'winter' time-series was collected, just that "early results in the time series were derived from only a few image pairs" (P16, L28-29), but the failure to capture the full winter velocity ensures that the subsequent trend analysis is flawed, especially given the dependence of the inter-annual trend analysis on this sample point at the very start of the time series five years prior to the next sample.**

*We fully acknowledged that these data are from a few-month periods and explicitly state that we are assuming these data representative of winter speeds. To evaluate the impact of this assumption, we compare several years of such radar winter estimates with GPS data and find good agreement (Figure 8). As we indicated for the comment above, any influence on the trend is likely to small.*

*Because the periods when the data were collected are published with the data online at NISDC, we made a point of not wasting valuable journal space with a table of dates. Given the concerns raised here, however, we will make it clear in the text that there are no September or May data in campaign data for the area covered by Figure 7 (there may be such data elsewhere on the ice sheet).*

*Given the numbers presented above and the good agreement with the GPS results shown in the text, we have no more "failed to capture the winter velocity" than T2015 have failed to capture the annual velocity by not using a sampling window of exactly one year, which, as shown above, can lead to larger biases than are present in our winter estimates.*

The sampling issue highlighted above is a more significant problem when considering relatively small absolute changes in velocity. The intra-winter velocity range of ~25-30 m/yr reported in Joughin et al. (2008) (and Colgan et al., 2012) is of the same magnitude as the overall annual velocity decrease reported in T2015; as such, any failure to correctly estimate winter motion in the present study will have considerable implications for a trend analysis.

> *This statement seems to be repeating earlier arguments – the 25-30 m/yr is a red herring as noted above. The GPS example above indicates small biases.*

The data underlying the J2018 trend analysis thus fail to capture net ice flow over annual and longer time scales, instead providing sub-annual snapshots of observation periods which vary in their acquisition time, both in length and period during the winter, from one winter to the next.

> *As indicated above any such biases are small, which we will augment the manuscript to indicate.*

They therefore have the potential to incorporate considerable variability in each of their derived 'winter' velocities, depending on the precise period of time that was sampled/available to derive their velocity estimate. We ask that the authors provide considerably more detail of their different winter 'snapshots', beyond the existing explanation at P16, L2-3 and without simply directing readers to the underlying NSIDC dataset metadata.

> *We will add clarification to the revision to provide more detail about the timing for this specific area, but we see no need to repeat metadata that are freely available (to the best of our knowledge the data presented in T2015 are not publically available).*

We note that J2018 provide a comparison between their radar derived NL data collected over "only a few months of each winter" and NL-GPS data reported in Stevens et al. (2016, GRL). They conclude that "most of the GPS points agree well with the radar-derived speeds" (P11, L17-18) and subsequently suggest that their "results are not unduly biased by seasonal variability" (P16, L5-6). We estimate, taking the data from Fig. 8., that while 'most' GPS do agree well, some comparisons are poor (e.g. 2008 where the SAR data looks to be ~10 m/yr too high, possibly due to seasonal bias).

> *In the above statement "some" actually means one. The point is about 9 m/yr faster than the GPS, or roughly 2 sigma (given the formal errors are not perfect this isn't bad). As noted above, ~2 m/yr of the difference might be due to seasonal bias, and the rest we expect is noise.*

While the comparison gives confidence that the radar is performing reasonably well in terms of absolute GPS displacement (+/- ~5 m/yr s.d.), such an error can introduce considerable variance in velocity trends when the trends are small in absolute terms.

> *Yes, but from what we can tell T2015 are estimating trends (Fig. 2) using 1-sigma errors ranging from about 5 to 15 m/yr and differences of maps mosaicked from results with errors of up to 60 m/yr, so why is that more valid than our results, which have considerably lower errors ? We also note that quite a bit of the area in the T2015 paper actually appears to have little or no change in speed (i.e., there is quite a lot of white area in Figure 1).*

As a result, the data as currently presented do not provide compelling evidence that InSAR is generating 'winter' velocities at the requisite temporal resolution that can ensure that the results "are not unduly biased by seasonal variability", especially when investigating trends in areas where the ice is moving slowly (~ 100 m/yr).

> *Would not such a statement also apply to the T2015 data, which have considerably larger errors and potential biases (as indicated above)? We feel the GPS results above (which we will fold into the revised discussion) provide a sound demonstration that our winter sampling is valid (the analysis is similar to the T2015, except we used actual velocities rather than idealized seasonal velocities in the T2015 paper, which can understate sampling biases).*

Last, J2018 criticise the T2015 choice of baseline period as opening the potential for seasonal variability to be aliased (J2016, P16, L6-9). However, we note that T2015 investigated this possibility in some detail (see Materials and Methods - 'Impact of varying baseline durations on annual velocity' and Fig. S1). J2018 do not make any reference to this analysis in their critique of T2015. To summarise, T2015 found that longer baseline periods beginning/ending in summer are likely to lead to a small artificial increase in ice motion, which is in the opposite direction to the decadal slowdown signal that is found and reported in the T2015 study area. In line with the T2015 baseline sensitivity analysis, we therefore ask that the authors demonstrate statistically that their own sub-sampling methodology has not impacted their results. Such an analysis should be robust for the whole SW sector analysed if the current conclusions are to be justified.

> *As requested we produced such analysis. We applied both our sampling strategy and the T2015 strategy to actual GPS data. The results suggest small trend biases for both, although such errors are 3x larger with the T2015 sampling strategy.*

(2) Comparison of the results presented in J2018 and T2015

In J2018 Fig. 7, the units are metres per year. If the aim of J2018 is to undertake a valid and direct comparison with T2015, the authors should use the same units,

namely percentage change relative to a reference period. In principle, we assume this would be 2000-01, although given the issues associated with the 'winter' sampling in 2000-1 (and the large error bars associated with this time period as shown in Fig. 8), this would likely be problematic because this earlier reference period may not be representative of net winter motion.

> *It is important to keep in mind that the purpose of this paper was not a redo of the T2015 results. Rather our purpose was to describe the data and demonstrate the quality of the data through several case studies (chosen for both demonstration purposes and scientific value). We also took a different, but equally valid perspective to the analysis (fitting trends to the data at each point, rather than a simple difference of two multi-year epochs).*

> **There is an overall trend presented in T2015 is -1.5 m/yr2 in Figure 2b, hence it is fully appropriate that we include our trends in these units. We did not provide a difference comparison to match that of T2015 because such a comparison as it beyond the scope of our paper (we also don't have 1985-94 base map to make such a comparison).**

On P11,L7-9, J2018 states that 'In the T2015 region (see black rectangle in Figure 7), we find some indication of slowdown, but the trends are less than those indicated by Tedstone et al. (2015)'. We request that the authors are more precise and provide, for example, an average velocity and
average % change for the region. This will ease comparison with both T2015 and the GPS measurements presented in the study. This is especially important given that J2018 find statistically significant evidence for a slowdown (Fig. 7 and P11, L7-9) but conclude later in the manuscript that it is due to aliasing of seasonal variability (P16, L6-7).

> *Our Figure 7 does present a speed map and one can reference the trend relative to that. With respect to a percentage change, T2015 differ an 80s/90s map with a 2007-14 map. Since we have no such prior basemap, it is not really possible to do such a comparison. What we said in the referenced section was "find some indication of slowdown, but the trends are less than those estimated by Tedstone et al. (2015)" What we said is simply a statement of fact (you might not believe our numbers, but we believe this to be an accurate statement of the differences). Our statement regarding trends was meant to be relative to the 1.5 m/yr2 value given in Figure 2. We will investigate revising the text to make this comparison clearer.*

The approach chosen for trend analysis (as opposed to differencing two time periods as per T2015) requires clearer explanation. For example, does the analysis take the formal velocity uncertainty at each pixel each year into account, and do they exclude potential outliers (robust linear regression)? What is the estimated error of the computed trends (i.e. computed from the covariance matrix)? Are the pixels used for

computing the trend analysis present in every single mosaic or are some pixels missing some time points? Furthermore, we note that a trend of 0 m/yr is a valid result that should not be excluded, yet it appears that these are not shown due to the filtering applied (Fig. 7 caption).

*Although there is some variation from year to year, the quality of all of the data is reasonably consistent over time. Thus, we did a simple linear regression with no weighting for errors (scipy.stans.linregress). Because there are some gaps in the data, we only computed fits were there were at least 6 points. Omission of the endpoints could have the biggest impact, especially for 2000. Only about 1% of these points have no 2000 data (mostly along the very far right of the region shown in figure 7) and there are valid data for all of the winter 2016 points (i.e., to the extent points are missing for some estimates, its mostly toward the middle of the interval where their omission should have the least effect)*

*To the extent possible, outliers have been rejected prior to the regression in the culling process that is part of the overall QA in creating the velocity maps.*

*Of the 27592 points where we evaluated trends, only 132 were rejected due to lack of data (we will amend the caption to indicate this point, but in general a lack of color in the figure means there was no statistically significant trend).*

*It's not a matter of us excluding statistically significant results of 0 m/yr (a trend that is not significantly different than 0 is indicated by a lack of color). We only excluded results that were statistically insignificant. Essentially if the null hypothesis is "that there is no trend in speed of the ice sheet," then the alternate hypothesis that there is a trend is validated once a trend is identified that meets the test for significance (we picked the very common p=0.05). As noted there are some 27000 pts, and a significance level of 0.05 implies if the null hypothesis were true, then we would expect 27592\*0.05=1379 points that incorrectly pass the test.  In fact, 4137 pass (~15%), but many of these occur on fast moving areas where a trend is expected (e.g., edge of Jakobshavn and glacier at the lower end of the box). Thus, if we exclude these areas where there are clear trends, the number that pass the test on much of the ice sheet area approaches what we would expect if there were no trend (or a trend for a small part of the area).*

*While a null hypothesis such as that above ("there is no trend in speed of the ice sheet") cannot be proven by an analysis such as ours, we can more conclusively state that "given the level of noise in our measurements, there is no detectable trend in speed of the ice sheet." It always possible that there is a weak trend that simply is not detectable. A long record or less noisy data might eventually reveal such a trend. That said, we do not feel that the T2015 data meet this criterion (slightly shorter post 2000 record, and larger uncertainty in the velocity data).*

We also note that the GPS observations presented show good agreement with T2015. We presume that, unlike the velocity estimates obtained from InSAR, the GPS dataset records net winter and annual ice motion as opposed to shorter temporal snapshots. NL-GPS (within the T2015 study area) has a computed slow-down of 1.3 m/yr (p=0.06) over the period 2007-2013 (Fig. 8), compared with the T2015 region-average of 1.5 m/yr during 2002-2014 (T2015 Fig. 2 and text). Meanwhile, J2018 fail to reproduce the GPS trend or T2015 trend with their InSAR observations (Fig. 8, NL timeseries). Similarly, a long-term decrease (1990-2012) in annual ice motion in this region has been measured by GPS – at the K-transect (van de Wal et al., 2015, The Cryosphere).

*The GPS observations show a short-period time series, so one has to be careful about saying it agrees with the T2015 (the T2015 data also show do not show much change at this elevation either). And in fact, in terms of the calculated trend, the InSAR data produce a very similar trend, albeit with no significance (p=0.36) – see Figure below.*

[Figure]

*This plot indicates two points:*

1) *Noise in the data can obscure a trend that less noisy data will find. For any given data set, the level at which a trend can be detected is noise dependent. While for satellite data it remains a challenge to detect such trends, we feel our data (and similarly derived insar results) offer the best performance. We certainly would not be surprised if longer and/or more accurate data reveal a long-term trend.*

*2)  As with many time series, it is easy to find a trend in a short section of the data that is not sustained over the long run (when we more than double the period the trend disappears). There are a variety of statistically insignificant signals in these data; for example, the GPS speeds increase from 2007 to 2009, then decrease afterwards.*

*We haven't analyzed the van de Wal data, but certainly their Figure 8 does not suggest much of slowdown trend from 2000 to 2012, which agrees well with our results for this time period. To the extent any slowdown is apparent, it is from 1990 to 2000, before any change in melt, which is in opposition to the T2015 finding.*

Last, we note that the discussion about summer ice motion at P16,L11-24 could be improved through stronger grounding in existing hydro-dynamic-coupling literature. For instance, studies such as Sole et al. (2013, GRL) show that summer velocities are faster than winter velocities, so proposing summer slow-down to below winter velocities as a possible explanation for T2015 but then immediately 'disproving' it (P16, L14-15) is confusing and has the potential to mis-lead. Similar datasets and discussion can be found in e.g. lead-authored work by Doyle, Sole, Bartholomew, Tedstone, Hoffman, Stevens. Moreover, given that the discussion makes comparison with T2015, it should also directly address T2015's hypothesis, namely that the processes responsible for the slow-down occur following the cessation of melting, i.e. early winter (e.g. T2015, p694, paragraph 1), *not* during summer.

*Such a discussion is beyond the scope of the text, and completely unnecessary as what we presented was a simple argument to answer the question "could the difference between winter an and annual velocities be explained by differences in summer velocities."  We simply say if this were the case, then what is the logical conclusion (a huge summer slowdown), which turns out to be completely inconsistent with observations. Hence, the difference cannot be explained by changes in the summer velocities, which our data do not sample. In fact, the last point above ("not during summer") is the exact point we were making (if it is in the annual, it has to be in the winter only too).*

*As noted above, such a literature review involves issues that are beyond the scope of the paper. For example, rather than supporting the hypothesis of the responsible processes, Stevens et al 2016 show that the correlation on which the hypothesis is founded is likely a statistical artifact : "Thus, the improved correlation observed when multiple years of runoff are included is an expected outcome of analyzing two variables with long-term temporal trends, even if the mechanism generating these trends is unrelated to the annual variability."*

*Summary: Both sets of data have issues with sampling, which we have tried to acknowledge in the original manuscript. Based on the comments, we note that in the revision we need to a) provide clarification on the sampling period b) to*

*fold in the analysis of the GPS data discussed above to make clear sampling biases are small, and c) to provide additional detail on how the trends were computed (i.e., how many points were used).*

*Many of the issues brought up in this comment have been shown to be unsubstantiated (for example, our analysis suggests our winter velocities should have an order of magnitude less bias than has been suggested, and in fact, less bias than the T2015 annual velocities).*

*We will add a sentence or two acknowledging that it is difficult to pull out such trends from data and we may be operating near the margin of what's achievable. That said, we note the errors in our data are considerably (order of magnitude) smaller than the T2015 velocity data, so we are inclined to believe that until shown otherwise by longer or cleaner data sets that there is no widespread trend in deceleration on the ice sheet in response to increased melting. To the extent that there may be some slowdown in the region examined by T2015, it does not extend other regions of the ice sheet with similar flow patterns and similar melt forcing. Thus, even if some slowdown occurs in their study region, it cannot be generalized to nearby regions with similar melt forcing (i.e., perhaps it's a localized influence – e.g., more water captured by the adjacent Jakobshavn basin, where slopes steepened greatly over the last decade).*

---

## Referee Comment (RC1) · Anonymous Referee #1 · 11 Apr 2018

This study builds on previous work by the authors to create a long-term record of ice flow on the Greenland Ice Sheet with a variety of satellite data. This manuscript describes new velocity products derived from Landsat-8 and Sentinel 1A/B, demonstrates that there is good agreement across platforms, and uses the extended velocity record to analyze velocity variations at different points on the ice sheet. The paper clearly demonstrates the importance of having a long, high resolution velocity time series for interpreting changes in Greenland. I thought the paper was well written and easy to follow. I have only a few comments.

p. 2, l. 9: Perhaps point out that Sentinel 1A/B are SAR satellites, since the previous paragraph mentions radar and optical imagery and to avoid confusion with Sentinel 2.

p. 3, l. 30: I would've liked to see a few statements about how Landsat-8 velocity

[Figure]

fields produced for the GIMP are different/better than others – or is it just that you use a control point procedure and previous studies didn't? Maybe then just say "... unlike in previous work, we use a control point procedure..." (I realize that this is discussed a bit in Joughin et al., 2017, but I still think a couple of sentences here would help the reader.)

Fig. 6: I don't understand what exactly is being plotted here. This needs some explanation. Is each data point the sum of all of the velocity pixels for a given glacier? Or are you sequentially adding the velocity time series somehow?

p. 11 / Fig. 7: Even if the trends in this data aren't as strong as reported by Tedstone et al., there does appear to be some slow down along the margins and some flow acceleration near the snowline. Can't that be taken as an indication of hydrologically driven changes? Or am I misunderstanding the figure? I would've like to have seen a little more detail regarding the comparison of these data with the figures from Tedstone et al., such as the average trend over the region and the percentage change. Anyway, it seems that the comments already posted to the manuscript discussion will help to clarify any confusions here.
* * *

---

## Referee Comment (RC2) · Anonymous Referee #2 · 23 Apr 2018

Review of: Greenland Ice Mapping Project: Ice Flow Velocity Variation at submonthly to decadal time scales

Joughin, Smith and Howat provide an assessment of new velocity data products being produced and distributed as part of the GIMP. They discuss the introduction of Sentinel 1a/b radar data into their datastream and the production of annual, winter, quarterly and monthly velocity products. There are no significant changes in processing methodologies from earlier documentation (Joughin, 2002; Joughin et al., 2010; 2017). The manuscript focuses on the agreement of Sentinel 1a/b with earlier results and the tradeoffs between temporal averaging and temporal resolution. The authors go on to demonstrate the scientific utility of their new dataset by: (1) contrasting sessional and interannual changes in flow for two major tidewater glaciers, (2) examine long-term

trends over an area of land terminating ice in south west Greenland and (3) describing commonalities in glacier change for large sectors of the ice sheet.

Overall opinion:

Overall the paper and respective data are valuable contributions to the glaciology community and The Cryosphere is an appropriate venue for this type of paper. I do however have relatively minor concerns regarding the scientific analysis and conclusions presented in the manuscript.

General comments:

Sentinel 1a/b

The description of how the data was processed is straightforward but I found the validation against other data sources lacking. The authors identify some areas of large disagreement between TSX and Sentinel 1 derived flow velocities (e.g. as shown in Figure 3) that they qualitatively suggest are largely a function of sensor resolution. It would be very helpful if the authors could provide a more extensive statistical validation / comparison between TSX and Sentinel 1 derived datasets.

Jakobshavn and Koge Bugt

The authors contrast recent changes in flow occurring for two major tidewater glaciers. While I personally found the comparison thought provoking, the conclusions seemed somewhat speculative. In particular the authors argue that changes in the terminus extent resulted in a large observed slowdown of the Jakobshavn glacier in 2017. This is not clearly agued from the data presented. Without a quantification of the uncertainties in ice thickness and bed elevation it is very difficult to discern the likelihood that changes in the terminus position contributed to the 2017 slowdown. Are the authors able to rule out changes in basal or lateral drag upstream of the terminus as a possible cause of the slowdown? What was the cause of the glacier advance during the winter of 2016 and why, unlike other years over the past decade, did it maintain floating tongue

in the summer of 2017. If the front of the glacier was indeed floating in 2017 I believe that the authors should be arguing that it is the position of the grounding line and not the terminus position that is modulating flow. While, as previously documented by the authors, there is a clear correlation between seasonal speedup and terminus position, the contribution of terminus position to the 2017 slowdown seems much more tenuous and likely requires more detailed study than provided here. I would suggest that the authors remove discussion of the role of terminus position on the 2017 slowdown or that they significantly expand their analysis to better support their assertions.

The discussion of bedrock erosion at Koge Bugt and the extrapolation to the "last stands" of glaciers in a warming world seemed to me to be tangential to the presented analysis and highly speculative. I would suggest that the authors remove this discussion or greatly expand their analysis to support their assertions.

Southwest Greenland Ice Sheet Trends

I have read the ongoing discussions between Tedstone et al. and the authors concerning the conclusions presented in this section and it seems to me that the disagreement largely stems from sampling periods, trend duration, and the definition of significance in trend. I would be very curious to see how well Figure 7 b agrees with the difference map presented in Tedstone et al. when a significance mask isn't applied. Also if the authors average over many points they could potentially increase the signal to noise allowing for the identification of subtle changes in grounded ice flow speed.

Specific Comments:

p3, l9-28 It would be highly valuable to provide an assessment of velocities generated using only the orbit vectors vs those generated using GCPs

p4, l5 It would be helpful if the "additional weightings" were listed here.

P7, l12 I would suggest applying formal error propagation to represent errors in velocity (I believe it's $ev = (ex*vx + ey*vy)/v$ instead of using the "combined standard deviation"

P11, l8 The trends shown in Tedstone et al. are presented as % total change from a reference velocity. From the results presented in Figure 7 it is not obvious that the results presented in the two papers differ significantly.

Figure 3: Not sure where distance starts from. Could line segment indicators of A – A' and B-B' be added to the inset.

---

## Short Comment (SC2) · 23 Apr 2018

**Reply to TC-2018-40-AC1 on 'Greenland Ice Mapping Project: Ice Flow Velocity Variation at sub-monthly to decadal time scales' by Joughin, Smith & Howat.**

A. Tedstone[1], P. Nienow[2], A. Dehecq[3] and N. Gourmelen[2]

[1] Bristol Glaciology Centre, University of Bristol, UK (previously at [2])

[2] School of GeoSciences, University of Edinburgh, UK

[3] NASA Jet Propulsion Laboratory, USA (previously at [2] and Université Savoie Mont-Blanc)

We thank the authors for their thorough response to our Short Comment dated 6 April 2018. We concur with their summary that 'both datasets have issues with sampling' and that 'we may be operating near the margin of what's achievable'. Broadly, we consider much of their final paragraph summary to be a welcome and nuanced discussion of their position, which we suggest could make an appropriate replacement to 'the trends Tedstone et al (2015) observe may be statistical artefacts resulting from some combination noise [sic] and a shorter-duration (after 2000) record'.

We do however wish to remark further on just one important caveat, which we suspect became lost in the length of our original comment. In brief: the baselines in this study versus T2015 are never going to be reconcilable in terms of the glaciological processes/flow regimes that they each capture.

For instance, if the processes responsible for the slowdown observed in T2015 occur primarily in the transition period from summer to winter (i.e. loosely late August through early October) then T2015 will capture the impact of these processes (albeit with the uncertainty introduced by the 352-400 day baseline) whereas J2018 with a focus on velocities observed during October-April will not. An example of how much variability occurs in this period compared to the rest of winter is quite visible in Colgan et al (2011, Fig. 13) and Joughin et al (2008, Fig 2) where we see, in the latter, a larger rise in velocity from ~DOY 235 to ~DOY 275 (difficult to identify precisely) compared to the subsequent winter period.

Furthermore, not only does the early autumn velocity minima vary between years but there are also considerable year-on-year variations in the precise overwinter velocity 'recovery' pattern both at and between sites (e.g. Colgan et al. 2011 and van de Wal et al 2015, Fig. 3). We therefore suggest that it is not the case that "If every one of our data sets were collected over the same 3-month period each winter, then the biases would have no effect on the trend since they would be approximately the same each winter".

In summary, for this study to identify a slowdown trend, the processes driving any slowdown would have to occur during the October to April sampling period, but there remains a significant likelihood that slowdown (or indeed speedup) processes occur outside these baselines and/or that comparisons using varying baseline periods may mask specific trends. We therefore hope that this caveat will be explicitly addressed in the revised manuscript – it doesn't invalidate either this study or T2015 but facilitates a more informed comparison to be made.

---

## Author Comment (AC2) · 1 May 2018

This document contains a point by point response to the reviewers and commenters, to which we have appended a change tracked version of the document.

Please also note the supplement to this comment:
https://www.the-cryosphere-discuss.net/tc-2018-40/tc-2018-40-AC2-supplement.pdf

---

## Author Response (AR1)

**Response to Reviewer 1**

**This study builds on previous work by the authors to create a long-term record of ice flow on the Greenland Ice Sheet with a variety of satellite data. This manuscript describes new velocity products derived from Landsat-8 and Sentinel 1A/B, demonstrates that there is good agreement across platforms, and uses the extended velocity record to analyze velocity variations at different points on the ice sheet. The paper clearly demonstrates the importance of having a long, high resolution velocity time series for interpreting changes in Greenland. I thought the paper was well written and easy to follow. I have only a few comments.**

> *No action required.*

**p. 2, l. 9: Perhaps point out that Sentinel 1A/B are SAR satellites, since the previous paragraph mentions radar and optical imagery and to avoid confusion with Sentinel 2.**

> *Added "SAR" between "Sentinel 1A" and "data"*

**p. 3, l. 30: I would've liked to see a few statements about how Landsat-8 velocity fields produced for the GIMP are different/better than others – or is it just that you use a control point procedure and previous studies didn't? Maybe then just say ". . . unlike in previous work, we use a control point procedure..." (I realize that this is discussed a bit in Joughin et al., 2017, but I still think a couple of sentences here would help the reader.)**

> *Tracking of Landsat-8 data is pretty straight forward and in principal our results should not be any better or worse than others. There probably at least a half dozen other LS8 and trying to track what each did is beyond the scope of this paper. We did add a bit more detail regarding the processing.*

**Fig. 6: I don't understand what exactly is being plotted here. This needs some explanation. Is each data point the sum of all of the velocity pixels for a given glacier? Or are you sequentially adding the velocity time series somehow?**

> *Updated the caption with "For these plots the bottom curve represents the speed of the first glacier (58 for the northwest). The next curve from the bottom is the sum of the first glacier and the second (58+56 for the northwest). Each successive curve is then the sum of the next glacier added to cumulative sum of the previous glaciers."*

**p. 11 / Fig. 7: Even if the trends in this data aren't as strong as reported by Tedstone et al., there does appear to be some slow down along the margins and some flow**

acceleration near the snowline. Can't that be taken as an indication of hydrologically driven changes? Or am I misunderstanding the figure?

> *It can be taken as evidence of change, which could be attributable to melt-intensity related hydrological processes, or other hydrological processes (water piracy being adjacent to a rapidly thinning basin), changes in geometry, or other possible causes. Or it could just be an artefact (e.g., a trend that will go away when the time series is extended). We have modified the text extensively both to demonstrate the limits of our trend detection and to augment the discussion of the agreement in terms of statistics.*

I would've like to have seen a little more detail regarding the comparison of these data with the figures from Tedstone et al., such as the average trend over the region and the percentage change. Anyway, it seems that the comments already posted to the manuscript discussion will help to clarify any confusions here.

> *It's important to note that our study is not meant to be a total redo T2015, but rather looking at whether there are trends over a broader area than they examined. We feel it makes the most sense to look at the raw trends, rather than normalizing them. Even if we did a percentage change, it would not be a direct comparison with T2015, since we compute trends fitted to multiple years and they compute differences of two multi-year averages, which correspond to a different period. As noted in the previous comment, we have done extensive revisions of this section, including Monte Carlo simulations to evaluate our detection threshold (figure S2).*

**Response to Reviewer 2**

Joughin, Smith and Howat provide an assessment of new velocity data products being produced and distributed as part of the GIMP. They discuss the introduction of Sentinel 1a/b radar data into their datastream and the production of annual, winter, quarterly and monthly velocity products. There are no significant changes in processing methodologies from earlier documentation (Joughin, 2002; Joughin et al., 2010; 2017). The manuscript focuses on the agreement of Sentinel 1a/b with earlier results and the tradeoffs between temporal averaging and temporal resolution. The authors go on to demonstrate the scientific utility of their new dataset by: (1) contrasting sessional and interannual changes in flow for two major tidewater glaciers, (2) examine long-term trends over an area of land terminating ice in south west Greenland and (3) describing commonalities in glacier change for large sectors of the ice sheet.

> *No action required.*

Overall opinion: Overall the paper and respective data are valuable contributions to the glaciology community and The Cryosphere is an appropriate venue for this type

of paper. I do however have relatively minor concerns regarding the scientific analysis and conclusions presented in the manuscript.

*See response below.*

General comments:

**Sentinel 1a/bL The description of how the data was processed is straightforward but I found the validation against other data sources lacking. The authors identify some areas of large disagreement between TSX and Sentinel 1 derived flow velocities (e.g. as shown in Figure 3) that they qualitatively suggest are largely a function of sensor resolution. It would be very helpful if the authors could provide a more extensive statistical validation / comparison between TSX and Sentinel 1 derived datasets.**

*There are so many ways to slice and dice the data that it's difficult to boil it down to a simple statement that TSX are this much better than Sentinel 1 data. For example, there are often more S1A/B data available, so if enough data are averaged the S1A/B data might be more accurate. The point we have tried to make is that in general while these data agree well, there will be places to watch out for (e.g., shear margins). Thus, we have tried to describe the types of errors users should be on the lookout for (resolution, geolocation, slope related). Figures 3–5 provide a demonstration of the types of differences one can expect to encounter.*

**Jakobshavn and Koge Bugt: The authors contrast recent changes in flow occurring for two major tidewater glaciers. While I personally found the comparison thought provoking, the conclusions seemed somewhat speculative. In particular the authors argue that changes in the terminus extent resulted in a large observed slowdown of the Jakobshavn glacier in 2017. This is not clearly agued from the data presented. Without a quantification of the uncertainties in ice thickness and bed elevation it is very difficult to discern the likelihood that changes in the terminus position contributed to the 2017 slowdown. Are the authors able to rule out changes in basal or lateral drag upstream of the terminus as a possible cause of the slowdown?**

*Much of the analysis just described has already been applied to the earlier record using a variety of models, hence we base our conclusion on the results from the earlier work, which do demonstrate there is sufficient sensitivity to explain the new observations. Lateral drag and basal drag are addressed in more detail in Joughin et al, 2012.*

*Our discussion here is more qualitative than quantitative, but the magnitudes of the change are well with the range of the previously established sensitivity to terminus depth/thickness. We kept it qualitative precisely because there is some degree of uncertainty in the bed. We have extensively reworded the text in this section to make the points more clear.*

**What was the cause of the glacier advance during the winter of 2016 and why, unlike other years over the past decade, did it maintain floating tongue in the summer of 2017. If the front of the glacier was indeed floating in 2017 I believe that the authors should be arguing that it is the position of the grounding line and not the terminus position that is modulating flow. While, as previously documented by the authors, there is a clear correlation between seasonal speedup and terminus position, the contribution of terminus position to the 2017 slowdown seems much more tenuous and likely requires more detailed study than provided here. I would suggest that the authors remove discussion of the role of terminus position on the 2017 slowdown or that they significantly expand their analysis to better support their assertions.**

> *We have substantially revamped the text here to better explain our arguments, but the statements here are supported by extensive model-based analysis in Joughin et al, 2012.*

**The discussion of bedrock erosion at Koge Bugt and the extrapolation to the "last stands" of glaciers in a warming world seemed to me to be tangential to the presented analysis and highly speculative. I would suggest that the authors remove this discussion or greatly expand their analysis to support their assertions.**

> *The text is speculative in that we are putting an idea out there that is suggested by the data but would require work well beyond the scope of this paper to prove. But much of science is about putting such ideas or hypotheses forward, so that they can later be proved or disproved. It is not the main thrust of the paper and the text is clearly written to indicate we are hypothesizing, which is entirely appropriate in a discussion section. That said, we changed a few words (e.g., "likely" to "may") to make it clear that this is speculation directed at forming a hypothesis, which is a completely legitimate and an essential part of science.*

**Southwest Greenland Ice Sheet Trends: I have read the ongoing discussions between Tedstone et al. and the authors concerning the conclusions presented in this section and it seems to me that the disagreement largely stems from sampling periods, trend duration, and the definition of significance in trend. I would be very curious to see how well Figure 7 b agrees with the difference map presented in Tedstone et al. when a significance mask isn't applied. Also if the authors average over many points they could potentially increase the signal to noise allowing for the identification of subtle changes in grounded ice flow speed.**

> *As noted in responses to other comments, the we compute trends over 16 year period (2000/1 to 20016/17) whereas T2015 difference decadal scale maps (1985-1994 and 2007-2014). Extensive revisions have been made. In particular, the analysis of the limits on detection (see text and new Figure S2).*

Specific Comments:

**p3, l9-28 It would be highly valuable to provide an assessment of velocities generated using only the orbit vectors vs those generated using GCPs**

*Agreed, and at some point, we plan to adapt our processing chain to use either approach, but that is beyond the scope of the present paper.*

**p4, l5 It would be helpful if the "additional weightings" were listed here.**

*This sentence begins "As described below…" The additional weighting is somewhat complicated and is described in detail in the following paragraphs, which is why we put the pointer to it. It would confuse the issue to boil down those paragraphs into a few words to list the weightings in this paragraph.*

**P7, l12 I would suggest applying formal error propagation to represent errors in velocity (I believe it's ev = (ex\*vx + ey\*vy)/v instead of using the "combined standard deviation"**

*Instead we have updated the plot to show the individual standard deviations for the x and y components of velocity, which clearly demonstrates the asymmetry between components.*

**P11, l8 The trends shown in Tedstone et al. are presented as % total change from a reference velocity. From the results presented in Figure 7 it is not obvious that the results presented in the two papers differ significantly.**

*What has somewhat become lost in all of the discussion is that this was never meant to be a direct comparison to T2015. The larger point was when we look at a much bigger portion of the coastline, we see little in the way of any kind of sustained slow down. See comments above regarding percentage. We have extensively revised the discussion of the comparison, including pointing out the differences aren't that large statistically.*

**Figure 3: Not sure where distance starts from. Could line segment indicators of A – A' and B-B' be added to the inset.**

*Done.*

**Response to second comment by Tedstone et al. (see separate response to first comment)**

We thank the authors for their thorough response to our Short Comment dated 6 April 2018. We concur with their summary that 'both datasets have issues with sampling' and that 'we may be operating near the margin of what's achievable'. Broadly, we consider much of their final paragraph summary to be a welcome and nuanced discussion of their position, which we suggest could make an appropriate replacement to 'the trends Tedstone et al (2015) observe may be statistical artefacts

resulting from some combination noise [sic] and a shorter-duration (after 2000) record'.

We do however wish to remark further on just one important caveat, which we suspect became lost in the length of our original comment. In brief: the baselines in this study versus T2015 are never going to be reconcilable in terms of the glaciological processes/flow regimes that they each capture.

For instance, if the processes responsible for the slowdown observed in T2015 occur primarily in the transition period from summer to winter (i.e. loosely late August through early October) then T2015 will capture the impact of these processes (albeit with the uncertainty introduced by the 352-400 day baseline) whereas J2018 with a focus on velocities observed during October-April will not. An example of how much variability occurs in this period compared to the rest of winter is quite visible in Colgan et al (2011, Fig. 13) and Joughin et al (2008, Fig 2) where we see, in the latter, a larger rise in velocity from ~DOY 235 to ~DOY 275 (difficult to identify precisely) compared to the subsequent winter period.

> *As we noted in the manuscript, it's hard to explain the difference due to the summer variability. It is even more difficult to explain in terms of changes in the Fall slowdown, the net effect of which is smaller than the summer speedup. In fact, our last two points do include the full September to May period. If the Fall slowdown were driving the change, then we expect these points to be biased much slower than the other points, which don't include the early Fall data, which would enhance a negative trend. But no such bias is observed. Our evaluation of the GPS data indicates we are looking at biases of at most about 2.6 m/yr.*

Furthermore, not only does the early autumn velocity minima vary between years but there are also considerable year-on-year variations in the precise overwinter velocity 'recovery' pattern both at and between sites (e.g. Colgan et al. 2011 and van de Wal et al 2015, Fig. 3). We therefore suggest that it is not the case that "If every one of our data sets were collected over the same 3-month period each winter, then the biases would have no effect on the trend since they would be approximately the same each winter".

> *Figure 3 of van de Wal does not demonstrate inter-annual variability. What it does show is that Fall slowdown drops the speed to about 10 to 15% below the annual average for a month or two (compared with the a 60%to 80% summer increase). To the extent that such interannual variability might be an issue, it is also an issue for the annual velocities, which could measure 1 or 2 fall slowdowns. The Colgan example is a bit extreme, because it varies from a period where there was no melt to one with lots of melt (most of the areas with slowdown in T2015 where well below this elevation so exposed to melted the whole period). I don't know what the annual averages of these data show, but*

*they seem more likely to indicate speedup than slow down. Also recall that the T2015 argument is that there is multi-year memory in the system, so this type of year-to-year variability should not skew the results.*

In summary, for this study to identify a slowdown trend, the processes driving any slowdown would have to occur during the October to April sampling period, but there remains a significant likelihood that slowdown (or indeed speedup) processes occur outside these baselines and/or that comparisons using varying baseline periods may mask specific trends. We therefore hope that this caveat will be explicitly addressed in the revised manuscript – it doesn't invalidate either this study or T2015 but facilitates a more informed comparison to be made.

*We have made several changes, including a more detailed analysis on our ability to detect trends. Based on this reasoning, at the 95% confidence level both data sets are probably not that much out of agreement. That said, the other parts of the ice sheet we examined do not have much detectable change. Which suggests that whatever caused slowdown in the T2015 may be not be generalizable to other regions with similar melt forcing (local effects such as the nearby drawdown of Jakobshavn Isbrae may play a role somehow).*

[revised manuscript text omitted]

Figure S2. Monte Carlo simulations to detect a trend in speed in the presence of noise. A synthetic velocity vector is used with $v_x$=100 m/yr in 2000, increasing by a prescribed trend in m/yr (x-axis) and sampled at each year where we have data. The y-component of velocity is assumed to be zero. To this trend, we add zero-mean Gaussian random noise ($\sigma_x$, $\sigma_y$). The mean errors for our data set are $\sigma_x = 2.9$ and $\sigma_y = 6.1$ m/yr, while the mean errors in along- and across- flow directions are 3.7 and 5.6 m/yr, respectively. The speed is most sensitive to noise directed along flow, so we have also included an example with x,y errors reversed, which in this case would be the worst case flow-direction for this level of error. For each trend value and noise level, we generated 10,000 realizations of synthetic data and counted how many time a non-zero trend would be detected when a value of p=0.05 is applied.

Moon, T., Joughin, I., Joughin, J., & Black, T. E. (2015). MEaSUREs Annual Greenland

Outlet Glacier Terminus Positions from SAR Mosaics, Version 1.

http://doi.org/http://dx.doi.org/10.5067/DC0MLBOCL3EL

---

## Author Response (AR2)

**1    Response to Reviewer 2**

**Greenland Ice Mapping Project: Ice Flow Velocity Variation at submonthly to decadal time scales**

**I find this paper rather challenging to review. One major issue is that the authors present detailed methods for the derivation of velocities from Sentinel-1a/b data but there is relatively little attention given to the corresponding methods and uncertainties for the analysis and discussion of observed changes in velocity and likely causes. To the author's credit, several of their arguments are partially or fully supported by previously published work. Nevertheless, in several cases, it is difficult for the reader to get a sense of the likelihood that the authors' conclusions are valid. Here I provide a few suggestions for further improvement:**

> *We are not sure what is meant by "valid" as we are using well-established methods. We address the more specific points individually below.*

**In the abstract the authors state "contrary to earlier results we do not find a slowdown over much of the southwest Greenland". The authors state in the manuscript that their results are likely not statistically different from those of Tedstone et al. and P11, L12 states that the authors do find a slowdown. I would therefore recommend removing/changing this statement from the abstract.**

> *This is a fair point and we have changed the abstract to read: "In addition, we computed trends in winter flow speed for much of the southwest margin of the ice sheet and find little in the way of statistically significant change over the period covered by our data."*

**-      I continue to recommend a thorough analysis of the agreement between TSX and Sentinel-1a/b velocities. What is the distribution of velocity differences for all TSX and S1 data separated in time by +/- 10 days? Can the authors plot the relationship between local velocity gradient and velocity difference? Can authors resample the TSX data to show the relationship between velocity gradients, image resolution and error? It would be very helpful to provide the results of the analysis as a table so that the users of the data have quantitative metrics to assess the new dataset.**

> *We are not really sure what the reviewer is looking for here. Not all of things asked for relate directly to TSX vs Sentinel 1 data – for example slope related effects would affect different TSX estimates different viewing geometries were used. In other words, TSX is generally better, but not a gold standard. The examples we have picked were specifically designed to reveal the types of problems people should watch out for. But for any difference between TSX and S1, there could be several causes – it's not nearly so simple as deriving a lookup table that says for*

*this slope or gradient of X the error will be Y. So we have not addressed this point beyond the original revision.*

**P10, L13-15: KB9 appears to have the smallest horizontal gradients in velocity but the maximum disagreement. This does not seem to support the conclusion that differences are a function of surface gradients and sensor resolution.**

*If we take the standard deviation of a typical TSX estimate for a box about KB9 that is similar in size to S1 resolution element, we get about +/- 200 m, most of which is related to the characteristics of the flow rather than velocity. So we could easily have systematic differences of the magnitude shown (ie. A single TSX point could be 200 m/yr or more greater than the average).*

**Section 3.5 and 3.6. As with Reviewer 1 I do not understand the stacked velocity plots or the % change numbers presented in the paper. What points are the velocities taken from?**

*We assume that the construction of the plot is understood on this round, since the caption provides a detailed description. We do see, however, that it's unclear where the points come from. We added "For each glacier, we sampled the centre of the main trunk at a point a few kilometres up stream of the terminus. We did not use a fixed distance because the position of many the termini vary secularly or seasonally by several kilometres over the period covered by the data set."*

**What does it mean if the cumulative velocity changes by X%...**

*It seems obvious, but we added. "(Percentage speedup is calculated as the change in cumulative speed.)"*

**Would it not make more sense to normalize by glacier velocity? This way the authors could make a statement along the lines of "on average glaciers sped up by X%.**

*No, it makes more sense to do it this way if you are trying to gage how discharge is changing. For example, if 9 really small glaciers sped up 100% but one really big one with discharge equal to all of the smaller ones stayed constant, then by normalizing we would say the average speedup was 90%, when in fact the total speedup was much smaller (roughly 50% if the 9 smaller glaciers flowed at 1000 m/yr and the fast one at 10,000 m/yr).*

**Where do these errors come from? They seem overly optimistic given the paucity of data before 2015. Do they agree well with the standard deviation to the residuals to the linear fits?**

*I can only assume the reviewer is talking about the southwest data since they weren't clear about the section here. We added to the caption "with error bars from the formal error*

*estimates." We refer the Reviewer to the methods section where describe these errors and reference papers where the estimates have been validated.*

- General grievance: All velocity axes are scaled to very large ranges which looks good but makes it difficult compare products. Since comparison of products is, to a large degree, the motivation for the figures, could the authors also provide a plot for slow moving glaciers (100 – 300 m/y) so that the reader can better assess the agreement between products?

*Yes, we agree, which is precisely why we dedicated one of our 3 case studies (southwest Greenland) to slow-moving ice (<100 m/yr). This example covers mosaicked products. We also refer the Reviewer to Figure 2, which shows the standard deviation for the whole ice sheet for the individual S1 estimates.*

*The selection of the fast glaciers reflects the fact these glaciers generally are of more interest.*

Speculations/conclusions of Section 4.2 rely heavily on estimates of basal/fjord topography. For increased transparency, can the authors kindly include the latest bed profile from bedmachine and discuss how using this profile would impact their conclusions.

*We would rather not do this. Basically, most of the bed DEMs show an overdeepening in this area, but some show it more on one side of the fjord than the other. For example, the An et al paper cited shows a bed profile that looks very much like the one we show, but when they plot their map with the Plummer DEM, the overdeepening goes away. And if we use our profile with their map, the overdeepening will disappear. We feel it's beyond the scope of this paper to do a detailed analysis of the various bed maps or try and show plots along differing profiles. We have made a point of being clear that there are differences.*

- The authors include a paragraph (P15, L26 to P16, L4) on why the Koge Bugt glacier may have stayed near its current position and why other glaciers will/have likely done the same. While not necessarily wrong, this paragraph has little to do with the rest of the analysis presented in the manuscript. Is this not a chicken or egg argument? Does the trough form where the ocean is or does the ocean go where the trough is? I don't think that anyone will dispute that abrasion is more prevalent where the ice enters the ocean and glaciers flow is at a maximum. But the location and rate of erosion will also depend on the geology and thermal regime. Since this study does not address geomorphologic processes by which glacier troughs are formed I continue to recommend that this paragraph be removed.

*As we noted before, the text makes it clear there that it's a hypothesis. While there may be some chicken and egg aspect, the data do reveal how a region of fast (5-10 k/yr) ice flow can be "pinned" to the head of the subglacial trough – which would not be immediately obvious*

*without the data to show that can happen. Calling out the potential implications is not unwarranted.*

**Minor Comments:**

**-        P5, L18: insert "is" between "it: and "best"**

*Done.*

**-        P13, L10: Where does the 1% value come from and is this not also the areas identified as most important and thus targeted by TSX?**

*It's a rough number – if you look at the percentage of the ice that flows greater than 500 to 1000 m/yr, you get a range of about 1.5 to .5% - we split the difference – 1% and put an ~.*

**A more quantitative analysis of the disagreements between sensors of different resolutions would help the reader to determine the impact of the observed difference.**

*We covered this issue above. As noted, there is not a clean way to quantify.*

**-        P14, L18 Why did the glacier advance in 2017 and not 2015 if it was colder in 2015?**

*There is not a one-to-one correlation, nor do we claim there is. But a period of extended cold winters may have contributed.*

**-        P15, L5 Is the distance that a perturbation propagates not also dependent on bed slope?**

*There are a number of potential factors and bed slope could be one as could be surface slope, and the sliding law and basal shear stress. Since we haven't ruled out these other causes, we changed "likely" to "may"*

**-        P16, S4.3 and 4.4. Why was Jakobshaven not included? The y-axis scales all the way to 80,000 m/yr... Jakobshaven would only increase this by another 14,000 m/y.**

*It could fit, but it is something of an outlier so we chose not to include it, especially since its treated separately.*

**-        P17, L17: Wouldn't a measurement that integrates over a much longer period be less sensitive to temporal sampling errors and also result in lower m/yr. errors? It seems a difference of 48 day would be inconsequential when looking at the total displacement over a full year. If velocities where 20% higher during the maximum period of sampling difference (48 days) then this would only result in a 2.6% error that should be random with time.**

*Basically, our sampling window is shorter, but it shifts within a period of relatively little temporal change.  While T2015 has a longer period, one measurement could for example include one summer peak with 2 fall periods with slowest velocities, while the second measurement it is compared with could include two summer peaks and one slow period. We did the quantitative analysis in the rebuttal from Tedstone et al (page 3) but didn't feel we need to provide an analysis of their data for this paper.*

- **P8, L31: Does the S1 processing use updated DEMs as done for the TSX processing?**

*No, but the period spanned is much shorter and we use the updated GIMP dem.*

- **P18, L2: elevation should be in units of "m"**

*Thanks, force of habit for someone who works with velocity. Fixed.*

[revised manuscript text omitted]